# Interplay between the EMT transcription factors ZEB1 and ZEB2 regulates hematopoietic stem and progenitor cell differentiation and hematopoietic lineage fidelity

Jueqiong Wang[1], Carlos Farkas[2,3], Aissa Benyoucef[2,3], Catherine Carmichael[1], Katharina Haigh[1,2,3], Nick Wong[1], Danny Huylebroeck[4,5], Marc P. Stemmler[6], Simone Brabletz[6], Thomas Brabletz[6], Christian M. Nefzger[7,8,9], Steven Goossens[10,11,12], Geert Berx[10,11], Jose M. Polo[6,7,8], Jody J. Haigh[1,2,3]*

1 Australian Centre for Blood Diseases, Monash University, Melbourne, Australia, 2 Department of Pharmacology and Therapeutics, Rady Faculty of Health Sciences, University of Manitoba, Winnipeg, Manitoba, Canada, 3 CancerCare Manitoba Research Institute, Winnipeg, Manitoba, Canada, 4 Department of Cell Biology, Erasmus University Medical Center, Rotterdam, the Netherlands, 5 Department of Development and Regeneration, KU Leuven, Leuven, Belgium, 6 Department of Experimental Medicine 1, Nikolaus-Fiebiger-Centre for Molecular Medicine, FAU University Erlangen-Nürnberg, Erlangen, Germany, 7 Department of Anatomy and Developmental Biology, Monash University, Melbourne, Australia, 8 Development and Stem Cells Program, Monash Biomedicine Discovery Institute, Melbourne, Australia, 9 Australian Regenerative Medicine Institute, Monash University, Melbourne, Australia, 10 Molecular and Cellular Oncology Laboratory, Department of Biomedical Molecular Biology, Ghent University, Ghent, Belgium, 11 Cancer Research Institute Ghent (CRIG), Ghent University, Ghent, Belgium, 12 Department of Diagnostic Sciences, Ghent University and University Hospital, Ghent, Belgium

* jody.haigh@umanitoba.ca

**Data Availability Statement:** Publicly available RNA-seq samples are currently available at NCBI BioProject PRJNA679880. All flow cytometry data

## Abstract

The ZEB2 transcription factor has been demonstrated to play important roles in hematopoiesis and leukemic transformation. ZEB1 is a close family member of ZEB2 but has remained more enigmatic concerning its roles in hematopoiesis. Here, we show using conditional loss-of-function approaches and bone marrow (BM) reconstitution experiments that ZEB1 plays a cell-autonomous role in hematopoietic lineage differentiation, particularly as a positive regulator of monocyte development in addition to its previously reported important role in T-cell differentiation. Analysis of existing single-cell (sc) RNA sequencing (RNA-seq) data of early hematopoiesis has revealed distinctive expression differences between *Zeb1* and *Zeb2* in hematopoietic stem and progenitor cell (HSPC) differentiation, with *Zeb2* being more highly and broadly expressed than *Zeb1* except at a key transition point (short-term HSC [ST-HSC]→MPP1), whereby *Zeb1* appears to be the dominantly expressed family member. Inducible genetic inactivation of both *Zeb1* and *Zeb2* using a tamoxifen-inducible Cre-mediated approach leads to acute BM failure at this transition point with increased long-term and short-term hematopoietic stem cell numbers and an accompanying decrease in all hematopoietic lineage differentiation. Bioinformatics analysis of RNA-seq data has revealed that ZEB2 acts predominantly as a transcriptional repressor involved in restraining mature hematopoietic lineage gene expression programs from being expressed too early in

generated and/or analyzed during the current study are available on Zenodo (DOI: 10.5281/zenodo. 5498282).

**Funding:** This work was partially supported by research grants to JJH from the National Health and Medical Research Council of Australia (GNT1104441, GNT1141081) and the Canadian Institutes for Health Research (166011). The funders had no role in study design, data collection and analysis, decision to publish, or preparation of the manuscript.

**Competing interests:** The authors have declared that no competing interests exist.

**Abbreviations:** AML, acute myeloid leukemia; BM, bone marrow; ChIP-seq, Chromatin Immunoprecipitation Sequencing; DEG, differentially expressed gene; DKO, double knockout; EMT, epithelial to mesenchymal transition; ETP-ALL, early thymic progenitor acute lymphocytic leukemia; FDR, false discovery rate; GO, gene ontology; HB, hemoglobin; HCT, hematocrit; HSPC, hematopoietic stem and progenitor cell; iDKO, inducible double knockout; KD, knockdown; KO, knockout; LSK, Lin⁻Sca1⁺cKit⁺; LT-HSC, long-term HSC; LYM, lymphocyte; MPP, multipotent progenitor; NEU, neutrophil; NK, natural; PB, peripheral blood; PCA, principle component analysis; PLT, platelet; RBC, red blood cell; RNA-seq, RNA-sequencing; ROS, reactive oxygen species; sc, single-cell; SPF, specific pathogen-free; ST-HSC, short-term HSC; WBC, white blood cell; wt, wild-type.

HSPCs. ZEB1 appears to fine-tune this repressive role during hematopoiesis to ensure hematopoietic lineage fidelity. Analysis of Rosa26 locus–based transgenic models has revealed that *Zeb1* as well as *Zeb2* cDNA-based overexpression within the hematopoietic system can drive extramedullary hematopoiesis/splenomegaly and enhance monocyte development. Finally, inactivation of *Zeb2* alone or *Zeb1/2* together was found to enhance survival in secondary MLL-AF9 acute myeloid leukemia (AML) models attesting to the onco-genic role of ZEB1/2 in AML.

## Introduction

Hematopoiesis is controlled by the tight coordinated interplay between the environment/ niche signals and cytokines, transcription factors, and epigenetic modulators to ensure lineage differentiation fidelity [1,2]. Alterations in these tightly controlled gene expression programs can lead to leukemia as well as other blood-related diseases [1,2]. Key transcription factors of the GFI (GFI1, GFI1b) and GATA (GATA1-3 of 6 members) families have been demonstrated to play essential roles in regulating hematopoietic stem and progenitor cell (HSPC) self-renewal, survival, as well as lineage-specific differentiation [3,4]. Each of the individual tran-scription factors of GFI and GATA family members have overlapping as well as distinct line-age-specific functions [3,4].

More recently, the ZEB family of transcription factors (ZEB1 and ZEB2) have emerged as key regulators of hematopoiesis and hematopoietic transformation [5,6] in addition to their roles in regulating epithelial to mesenchymal transition (EMT) processes in development and disease [7]. Hematopoietic-restricted knockout (KO) of *Zeb2* using Tie2 and Vav-iCre based approaches was found to result in multilineage differentiation defects, altered HSPC migra-tion, as well as embryonic/neonatal lethality, respectively [8]. An interferon-inducible Mx1-Cre based KO of *Zeb2* in the adult hematopoietic system resulted in increased granulo-cyte development, selective expansion of specific HSPC populations, and differentiation defects in erythroid/megakaryocyte as well as B cell lineages [9].

Full KO of *Zeb1* results in neonatal lethality due to multiple skeletal defects and associated breathing defects [10]. Surviving *Zeb1* null mice (20% of total KOs) showed defects in T-cell development [11] that progress to the development of a mature form of acute T-cell cutaneous leukemia on C57BL/6-C3H outbred backgrounds [12]. This disease resembles the T-cell defects observed in human Sézary syndrome that also display heterozygous and homozygous loss-of-function mutations in *ZEB1* [13,14]. Alternatively, overproduction of ZEB2 specifically in the murine hematopoietic system or in T cells can also selectively lead to T-cell transforma-tion, with mice developing an early block in T-cell development that resembles human early thymic progenitor acute lymphocytic leukemia (ETP-ALL), whereby patients also display up-regulated *ZEB2* [15]. These results suggest opposing roles for ZEB1 and ZEB2 in human T-cell development and transformation [6].

Within the myeloid lineage, however, there is emerging evidence that ZEB2 and ZEB1 may both potentially contribute to the development and/or maintenance of acute myeloid leukemia (AML) [16,17]. In MLL-driven forms of AML such as MLL-AF9 models, it has been demon-strated that both ZEB1 and ZEB2 may be direct transcriptional targets of this fusion protein [17,18] and may be essential downstream genetic determinants of AML progression and dis-ease severity.

Using both loss- and gain-of-function conditional Cre/*lox*P-dependent approaches, we have begun to further determine the role of *Zeb1* in hematopoietic differentiation alone and in

synergy with *Zeb2*. Here, we have found that hematopoietic-restricted KO of *Zeb1* results in cell-autonomous defects in hematopoiesis with clear defects in myeloid differentiation as well as loss of multilineage differentiation potential. Inducible KO of both *Zeb1* and *Zeb2* results in further blocks in hematopoiesis with dramatic increases in the number of HSPCs and such mice succumbing to lethal anemia/cytopenia within 2 weeks after tamoxifen-induced *Zeb1/2* double knockout (iDKO). These defects are selectively rescued through maintenance of a single wild-type (wt) *Zeb2* allele. RNA sequencing (RNA-seq)-based transcriptional analysis has found altered gene expression programs in *Zeb1/2* single and compound (double knockout, DKO) deficient Lin⁻Sca1⁺cKit⁺ (LSK) cells in genes involved in adhesion/migration, differentiation, stemness, as well as the inappropriate expression of immune and myeloid cell programs. Zeb2 appears to be dominant over Zeb1 as the major regulator of hematopoietic lineage fidelity.

Despite differential effects of *Zeb1* and *Zeb2* KO on myeloid differentiation, Rosa26 locus cDNA-based overexpression of either *Zeb1* or *Zeb2*, specifically within the hematopoietic system, was found to alter myeloid cell development equally leading to extramedullary hematopoiesis as well as monocytic lineage skewing. With regard to the role of Zeb1 and Zeb2 in AML progression, we have demonstrated that KO of *Zeb2* can significantly extend survival in MLL-AF9 AML transplant settings in vivo. Furthermore, KO of *Zeb1* in addition to *Zeb2* in MLL-AF9 settings did not further increase overall survival.

Overall, our results highlight both common as well as distinctive roles of ZEB1 and ZEB2 in regulating hematopoietic lineage fidelity and suggest similar effects on monocyte differentiation. Moreover, ZEB2 appears to be the dominant modulator of HSC multilineage differentiation, with ZEB1 fine-tuning this process.

## Results

### Expression analysis of *Zeb1* and *Zeb2* mRNA during early hematopoiesis

To determine unique as well as overlapping expression patterns for *Zeb1* and *Zeb2* mRNA in early murine hematopoiesis, we have reanalyzed the previously published [19] single-cell (sc) RNA-seq data with more recent bioinformatics tools [20,21]. In general, we discovered that irrespective of what mouse HSPC population is examined *Zeb2* appears to be relatively higher than *Zeb1* mRNA levels (**Fig 1A**). This is particularly true in the CMP/MEP(MMP2)/GMP (MMP3) populations. One exception to this trend is that *Zeb1* mRNA levels appear to be higher in short-term HSC (ST-HSC)/multipotent progenitor (MPP1) cells prior to commitment to further myeloid and lymphoid fate specification (MPP2 and MPP4/LMPP) (**Fig 1A**). In addition, a larger percentage of cells within these CMP/MEP(MMP2)/GMP(MMP3) populations express *Zeb2* than *Zeb1* (heatmaps of sublineage frequencies in **Fig 1A**).

Using sc mapping and pseudotime algorithms [20,21] we can lineage trace long-term HSC (LT-HSC)➔HSPC➔Prog cell transitions with LT-HSCs (occupying the left most part of graphs) and the most differentiated GMP committed progenitors (on the far right; **Fig 1B**). As expected, we discovered a major cluster containing LT-HSC and HSPC in the beginning of the pseudotime that drives differentiation of more committed MMP (HSPC) and progenitors (CMP /MEP/GMP) in a single trajectory, but in parallel also gives rise to a small population of LT-HSCs. We hypothesized this cluster will serve as a self-renewal niche rather than a source for differentiation, due to its position in the terminal branches of the trajectory curve [22] (clusters 3 and 4 in **Fig 1B**). According to more continuous differentiation models of hematopoiesis [23], these populations readily express some myeloid as well B and T-cell markers not seen in LT-HSCs and may represent more lineage primed HSPCs.

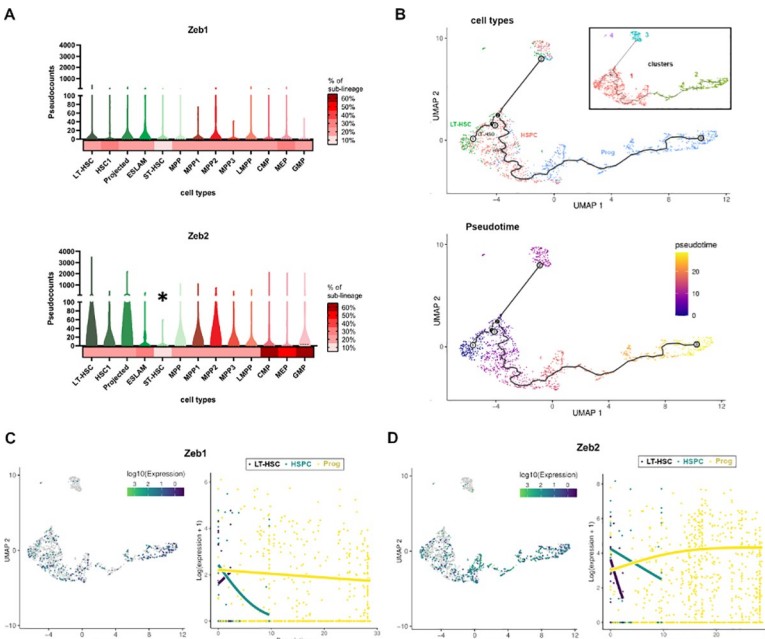

**Fig 1. Expression analysis of Zeb1 and Zeb2 during early hematopoiesis. (A)** Levels of *Zeb1* (upper) and *Zeb2* (lower) across LT-HSC (dark green to light green channels, Lin⁻c-Kit⁺ Sca1⁺CD34⁻Flk2⁻), HSPC (dark red to purple, LSK) and progenitors (dark red to light red channels, LSK), respectively. *Zeb2* levels are higher throughout except for ST-HSCs where *Zeb1* levels are higher (asterisk). Heatmaps at the bottom of graphs show percentage of cell lineage that express *Zeb1* or *Zeb2*. HSPC subpopulations were subdivided into HSC1 and HSC2 (projected) expressing low/absent and high SLAM marker CD229, respectively, including ESLAM cells (CD45⁺EPCR⁺CD48⁻CD150⁺). **(B)** (Upper) dimensionality reduction and trajectories of 3 major cell types (LT-HSC, HSPC, and Prog) across 1,920 hematopoietic single cells sequenced by Nestorowa and colleagues [19], analyzed with the Monocle3 algorithm [20]. Clusters identified by Monocle3 were enclosed in a black box. Two major clusters consisting in LT-HSC and HSPC (see clusters 1 and 3, respectively) and 1 cluster of progenitors (see cluster 2) were detected. Also, another restricted cluster of LT-HSC were found (see cluster 4). (Lower) Pseudotime fitting analysis of these cells with tradeSeq program [21]. **(C)** (Left) Expression of *Zeb1* gene across the single cell experiment, color coded according to the log10 of *Zeb1* pseudocounts. Cells with zero pseudocounts for *Zeb1* expression are colored in gray. (Right) Estimated expression obtained for *Zeb1* across cell types expressing *Zeb1* and arranged by pseudotime with the tradeSeq program. Across pseudotime, fitted curves indicates up-regulation of *Zeb1* expression in LT-HSC (black curve), down-regulation of *Zeb1* in HSPC (green curve) and mild down-regulation of *Zeb1* in progenitors (yellow curve). **(D)** (Left) Same as left C for *Zeb2* gene. (Right) Same as right C for *Zeb2* gene. Across pseudotime, fitted curves indicate up-regulation of *Zeb2* expression in LT-HSC and HSPC (black and green lines, respectively) and up-regulation of *Zeb2* in progenitors (yellow curve). HSPC, hematopoietic stem and progenitor cell; LSK, Lin⁻Sca1⁺cKit⁺; LT-HSC, long-term HSC; ST-HSC, short-term HSC.

Using these data, we could then plot *Zeb1* and *Zeb2* expression data versus pseudotime to understand better how *Zeb1* and *Zeb2* mRNA steady-state level changes are occurring throughout early hematopoietic development. Here, we could show that (1) *Zeb2* mRNA levels start out high in LT-HSC and HSPC cells, then gradually fall off, but then start to steadily increase as lineage progenitors become more committed (**Fig 1C**); (2) *Zeb1* mRNA levels increase as LT-HSCs differentiate, but then fall off quickly in HSPCs, and then gradually decrease in more committed progenitors; and (3) in addition to being more highly expressed, many more progenitors express *Zeb2* than *Zeb1* transcripts (**Fig 1D**).

From the ImmGen gene expression database (https://www.immgen.org/), there appears to be overlapping and, in some cases, complementary expression between *Zeb1* and *Zeb2*, with *Zeb1* being more highly expressed in mature B/T lymphoid cell subsets and *Zeb2* more highly expressed in the monocytic/macrophage lineages (**S1 Fig**).

## Hematopoietic-restricted genetic inactivation of *Zeb1* does not lead to excessive embryonic/postnatal lethality or cephalic hemorrhage

In order to study ZEB1 in embryonic hematopoiesis, we genetically intercrossed *Zeb1* conditional mice (*Zeb1*$^{fl/fl}$) [24] with the Tie2-Cre [25] and hematopoietic-restricted Vav-iCre [26] lines, as previously performed for *Zeb2* (using *Zeb12*$^{fl/fl}$) [8]. Despite the fact that Tie2-Cre$^{Tg/+}$; *Zeb1*$^{fl/fl}$ and Vav-iCre$^{Tg/+}$; *Zeb1*$^{fl/fl}$ mice were present at sub-mendelian rates at weaning (7% Tie2-Cre$^{Tg/+}$; *Zeb1*$^{fl/fl}$ versus 12.5% control Cre -ve; *Zeb1*$^{fl/fl}$ and 20% Vav-iCre$^{Tg/+}$; *Zeb1*$^{fl/fl}$ versus 25% Cre–ve; *Zeb1*$^{fl/fl}$), we observed no evidence for embryonic or postnatal lethality associated with cephalic hemorrhaging, which was observed for such *Zeb2*-deficient models [8] (**S2A Fig**): There, no viable embryos were observed at E13.5 (with Tie2-Cre), and only 3% of the mutant homozygous null pups survived at P7 (with Vav-iCre) [8] (**S2B Fig**). One major difference here is that the *Zeb1* KO experiments were performed on a pure C57Bl/6 inbred background, whereas the original *Zeb2* KO experiments [8] were originally performed on a mixed outbred background.

## Hematopoietic-restricted KO of *Zeb1* leads to decreased HSPC populations, decreased myeloid cell development, and altered T-cell differentiation

To document the cell-autonomous role of Zeb1 in definitive hematopoiesis, CD45.2$^+$ HSPCs were isolated from E14.5 fetal livers from Tie2-Cre; *Zeb1*$^{fl/fl}$ embryos and used to reconstitute lethally irradiated syngeneic C57Bl/6 CD45.1 mice in long-term reconstitution assays (**Fig 2A**). At 23 weeks post-reconstitution, HSPC analysis (**Fig 2B**) showed no changes in % or total number of SLAM marker (CD150$^+$CD48$^-$) LT-HSCs (**Fig 2C**, **S3A Fig**). Overall, there were decreases in % and cell number of Lin$^-$Sca$^+$cKit$^+$ cells (**Fig 2D**, **S3B Fig**: *$p < 0.05$). CD135/ CD34 staining of this Lin$^-$Sca$^+$cKit$^+$ population showed decreases in % of live ST-HSCs and MPPs but no significant changes in % LT-HSCs or in overall cell numbers (**Fig 2D**, **S3B Fig**: *$p < 0.05$, ***$p < 0.001$). FCγR$_{II/III}$/CD16-CD34 analysis of the MPP population (Lin$^-$cK-it$^+$Sca1$^-$) showed significant decreases in % of CMP (CD34$^+$FCγR$_{II/III}$$^{low}$) and GMPs (CD34$^+$FCγR$_{II/III}$$^+$), but no changes in MEPs (CD34$^-$FCγR$_{II/III}$$^-$) as well as overall decreases in GMP numbers (**Fig 2D**, **S3B Fig**: *$p < 0.05$, ***$p < 0.001$). Moreover, decreases in CD11b$^+$ myeloid cells (particularly monocytic lineage cells (i.e., CD11b$^+$, LyS6G$^-$) were observed in the peripheral blood (PB) and bone marrow (BM) as well as decreased neutrophils (NEUs; i.e., CD11b$^+$, LyS6G$^+$) in the BM (**Fig 2E**, **S3C and S3D Fig**: *$p < 0.05$, ****$p < 0.0001$).

Given the previous defects observed in T-cell differentiation in *Zeb1* global deletion mice [11], we also examined T-cell development in these transplant settings after long-term reconstitution. We could demonstrate significantly decreased thymocyte numbers with significant decreases mainly in % of live DN4 cells (i.e., CD25$^-$, CD44$^-$) as well as increased CD8$^+$ skewing in the thymus (**Fig 2F**: **$p < 0.01$, ***$p < 0.001$). No significant changes were observed in overall B cell differentiation (B220 marker analysis; **S3E Fig**).

Overall, these results are slightly in contrast to hematopoietic *Zeb2* null mice that showed enhanced granulocytic differentiation, as well as increased HSPC populations including MEPs, suggesting potentially divergent roles of Zeb1 and Zeb2 in controlling multilineage hematopoietic differentiation [9].

## *Zeb1* null HSPCs show decreased self-renewal and multilineage differentiation potential

To examine the cell-autonomous role of ZEB1 in HSC maintenance and self-renewal, we performed serial methylcellulose replating assays of *Zeb1*-deficient HSPCs obtained from E14.5

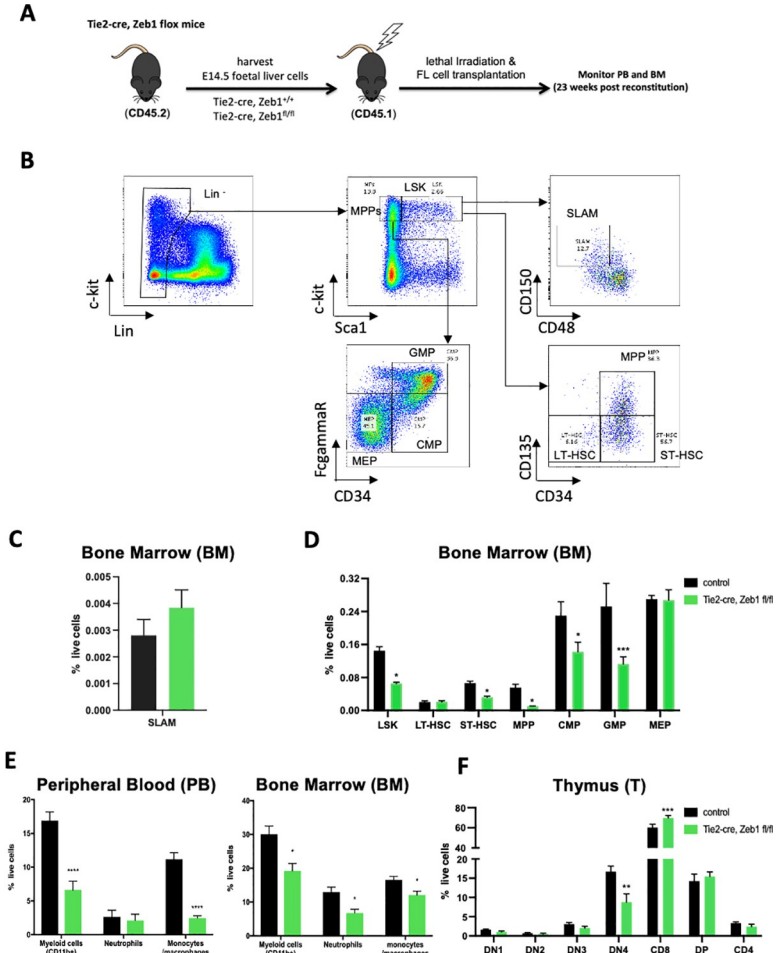

**Fig 2. Hematopoietic-specific loss of Zeb1 leads to differentiation defects in specific HSPC populations as well as myeloid lineage and T-cell defects.** (A) Schematic of BM transplant experiments using Zeb1 null CD45.2$^+$ fetal liver HSPCs from Tie2-Cre, *Zeb1$^{fl/fl}$* mice, and Cre only controls transplanted into CD45.1$^+$ recipients with PB and BM analysis conducted at 23 weeks post-transplant. (B) Overview of flow cytometry gating strategy used to analyze hematopoietic stem and progenitor (HSPC) populations. LSK cells were analyzed for SLAM marker expression (CD150, CD48) or were analyzed by parallel (CD135/CD34) marker expression to define MPP as well as ST and LT HSC populations more accurately. MPPs were analyzed by FcgammaR, CD34 expression to further define MEP, GMP, and CMP populations. (C) SLAM marker expression showing similar percentage of LT-HSCs (CD150$^+$CD48$^−$) in Zeb1 null and control BM. (D) Overall, there were decreases in the percentage of Lin$^−$Sca$^+$cKit$^+$ cells likely composed of significant decreases in the percentage *Zeb1*-deficient BM percentages for ST-HSCs and MPPs. Moreover, there were significant decreases in the percentage of as CMP and GMP cell populations in the BM of Zeb1 null reconstituted recipients. (E) Flow cytometric analysis of PB (left panel), BM of reconstituted mice showed defects in *Zeb1* null HSPC contribution to myeloid cells (Cd11b$^+$) including monocytic (Cd11b$^+$Ly6G) and NEU (Cd11b$^+$Ly6G$^+$) lineage cells. (F) Cytometric analysis of thymic T-cell populations showed significantly decreased percentage of CD25$^−$CD44$^−$ DN4 progenitors and increased CD8$^+$ mature T cells. Error bars indicate SD of the mean (*n* = 4 per group, $^*p < 0.05$, $^{**}p < 0.01$, $^{****}p < 0.0001$, nonparametric *t* test). Raw data behind graphs are included in A in S1 Data. BM, bone marrow; HSPC, hematopoietic stem and progenitor cell; LSK, Lin$^−$Sca1$^+$cKit$^+$; LT-HSC, long-term HSC; MPP, multipotent progenitor; NEU, neutrophil; PB, peripheral blood; ST-HSC, short-term HSC.

fetal livers of Vav-iCre; *Zeb1$^{fl/fl}$* embryos as well as Cre-negative controls. We have previously demonstrated that the Vav-iCre line is specific to the hematopoietic lineage at this stage, whereas the Tie2-Cre system is active in both the endothelium and hematopoietic system [8]. Here, we demonstrated decreased hematopoietic colony formation potential of *Zeb1*-deficient HSPCs; in primary plating experiments there was a 50% reduction in myeloid lineage colony

numbers (**Fig 3A**). This decrease in colony numbers was exacerbated in secondary plating experiments whereby the *Zeb1*-deficient HSPCs gave rise to 80% to 90% fewer myeloid lineage colonies compared to controls (**Fig 3A**: $^*p < 0.05$, $^{***}p < 0.001$).

To further demonstrate that *Zeb1* null HSPCs are deficient in self-renewal and multilineage differentiation potential, we performed competitive BM repopulation assays. Here, equal numbers of BM derived HSPCs from Vav-iCre$^{Tg/+}$; *Zeb1* $^{fl/fl}$ homozygous, Vav-iCre$^{Tg/+}$; *Zeb1*$^{fl/+}$ heterozygous and Cre-negative *Zeb1*$^{fl/fl}$ CD45.2$^+$ mice were mixed with equal numbers of competitor HSPCs from wt CD45.1$^+$ BM. These BM mixtures were used to reconstitute lethally irradiated CD45.1$^+$ recipient mice (**Fig 3B**). CD45 cytometric analysis of the PB and BM of long-term reconstituted BM recipients clearly demonstrated that *Zeb1*-deficient donor HSPCs (CD45.2$^+$) were outcompeted by wt competitor cells (CD45.1) in their capacity to give rise to all mature hematopoietic cells including B, T-, as well as myeloid cells in the PB and BM (**Fig 3C**). The overall percentages of chimerism and lineage-specific chimerism for these experiments are summarized in A–C of **S1 Table**.

These results are similar to the results obtained from the previous analysis of *Zeb2*-deficient HSPCs in competitive reconstitution experiments: These demonstrated that *Zeb2* null HSPCs could be outcompeted by wt competitor cells for their contribution to all mature hematopoietic cells, with the exception of granulocytes where there was some contribution of *Zeb2*-deficient CD45.2$^+$ progenitor cells [9].

## DKO of *Zeb1* and *Zeb2* causes PB cytopenia and severe differentiation defects in HSPCs

Given this similarity in *Zeb2*- and *Zeb1* null hematopoietic cells in being outcompeted by wt control HSPCs for their ability to contribute to multilineage hematopoiesis, we next examined if inducible loss of both Zeb1 and Zeb2 may exacerbate the severity of observed hematopoietic phenotypes of the individual gene KOs. For this, we used the tamoxifen-inducible Rosa26-Cre-ERT2 [27] system that was bred into conditional *Zeb1*$^{fl/fl}$; *Zeb2*$^{fl/fl}$ [28] and compound *Zeb1*$^{fl/fl}$; *Zeb2*$^{fl/fl}$ backgrounds (**Fig 4A**). To validate this inducible system, we used R26-Cre-ERT2; *Zeb2*$^{fl/fl}$ and R26-Cre-ERT2; *Zeb1*$^{fl/fl}$ BM cells in transplant experiments into lethally irradiated syngeneic hosts and allowed recovery for up to several weeks prior to tamoxifen-inducible KO of *Zeb2* or *Zeb1* alone. Induced Cre in this system lead to *Zeb2* hematopoietic deficient mice that phenocopied the original *Zeb2* phenotypes observed using the interferon-inducible Mx1-Cre model. These defects included excessive granulocyte differentiation, decreased monocyte, and B cell differentiation, as well as defects in erythroid and megakaryocytic differentiation, at 8 weeks after inducing the *Zeb2* KO [9] (**S4 Fig**).

We performed similar *Zeb1* KO experiments with the Rosa26-Cre-ERT2 line (**S5A Fig**) and could confirm efficient inactivation of the conditional *Zeb1* allele in whole BM cell populations (**S5B Fig**). Moreover, we confirmed the cell-autonomous defects of Zeb1-deficient HSPCs in methylcellulose replating experiments that we observed using the constitutive Vav-iCre model. Further, we demonstrated that tamoxifen-mediated KO of *Zeb1* in HSPCs severely compromised their ability to give rise to hematopoietic colonies that was already present at initial plating. However, this became sequentially worse in terms of number of colonies generated (compared to controls) by the second replating experiments (**S5C Fig**: $^{**}p < 0.01$). Finally, analysis of T-cell contribution of Zeb1-deficient adult HSPCs from this inducible model also showed similar T-cell defects as those documented in the constitutive Vav-iCre model, with decreased DN4$^+$ cells, but, here, we observed increased skewing to CD4$^+$ T-cell development (**S5D and S5E Fig**: $^*p < 0.05$, $^{***}p < 0.001$).

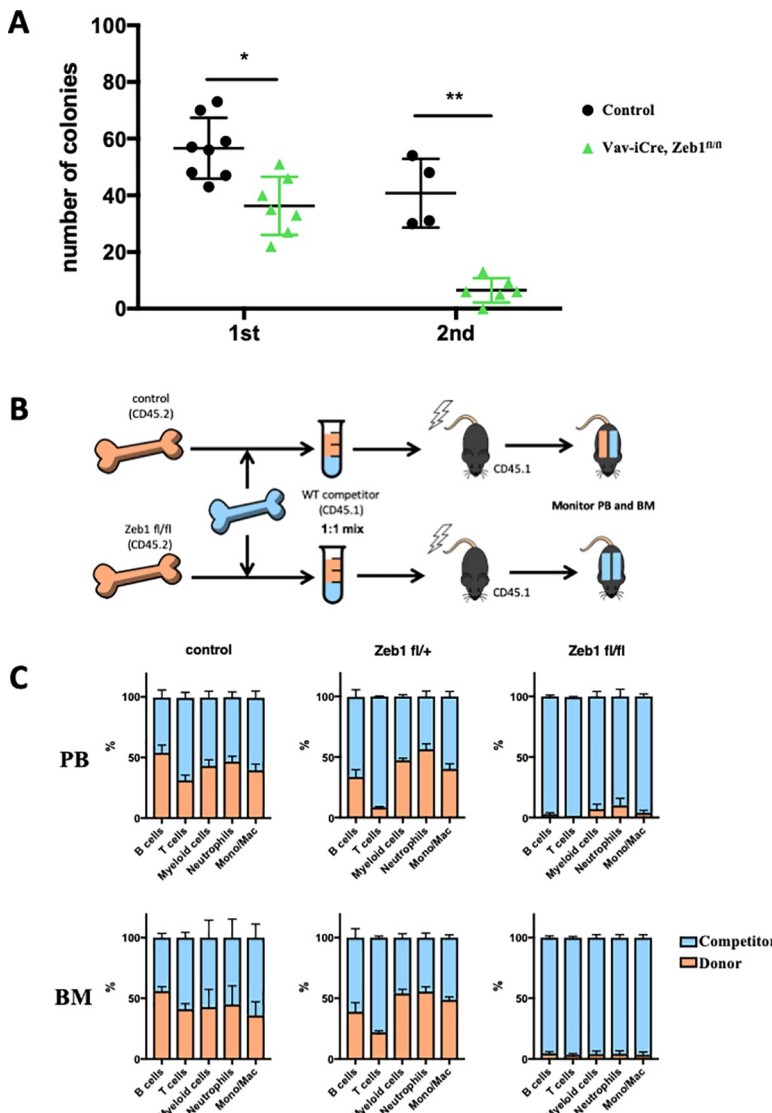

**Fig 3. Hematopoietic-restricted loss of Zeb1 decreases hematopoietic colony formation potential and ability to compete with wt HSPCs for contribution to all hematopoietic lineages. (A)** HSPCs isolated from Vav-iCre; *Zeb1*$^{fl/fl}$ fetal livers show decreased numbers of colonies in methylcellulose-based colony assays (first) that further decreases upon secondary replating (second) compared to Cre negative controls. **(B)** Schematic of competitive BM reconstitution experiments whereby equal numbers of BM cells from *Zeb1* null CD45.2$^{+}$ mice were mixed with equal numbers of control CD45.1$^{+}$ cells and used to reconstitute lethally irradiated CD45.1$^{+}$ mice. If *Zeb1* null CD45.2 + HSPCs are not compromised, they would be expected to contribute equally as the control CD45.1+ cells in their contribution to all hematopoietic cells (equal CD45.2$^{+}$-orange/CD45.1$^{+}$-blue, top row) whereas if they are severely compromised then the control CD45.1$^{+}$ cells will solely contribute to the reconstituted hematopoietic system (all CD45.1$^{+}$- blue, lower row). **(C)** *Zeb1* null (Vav-iCre, *Zeb1*$^{fl/fl}$) CD45.2$^{+}$ donor cells (orange bars) were outcompeted by control CD45.1$^{+}$ competitor HSPCs (blue bars) for their ability to contribute to all hematopoietic cells analyzed in the PB and BM (rightmost panels). *Zeb1*$^{fl/+}$ heterozygous (middle panels) and Cre negative (left panels) CD45.2$^{+}$ doner cells in general contributed equally as well as the competitor CD45.1$^{+}$ cells for their contribution to the hematopoietic system of recipient mice with the exception to the T-cell lineage. Data are represented as mean + SD from 4 biological replicates. $^{*}p < 0.05$; $^{**}p < 0.01$, nonparametric *t* test. Raw data for (A) are included in S1 Data. Raw data for (C) are included in B of **S1 Table**. BM, bone marrow; HSPC, hematopoietic stem and progenitor cell; PB, peripheral blood; wt, wild-type.

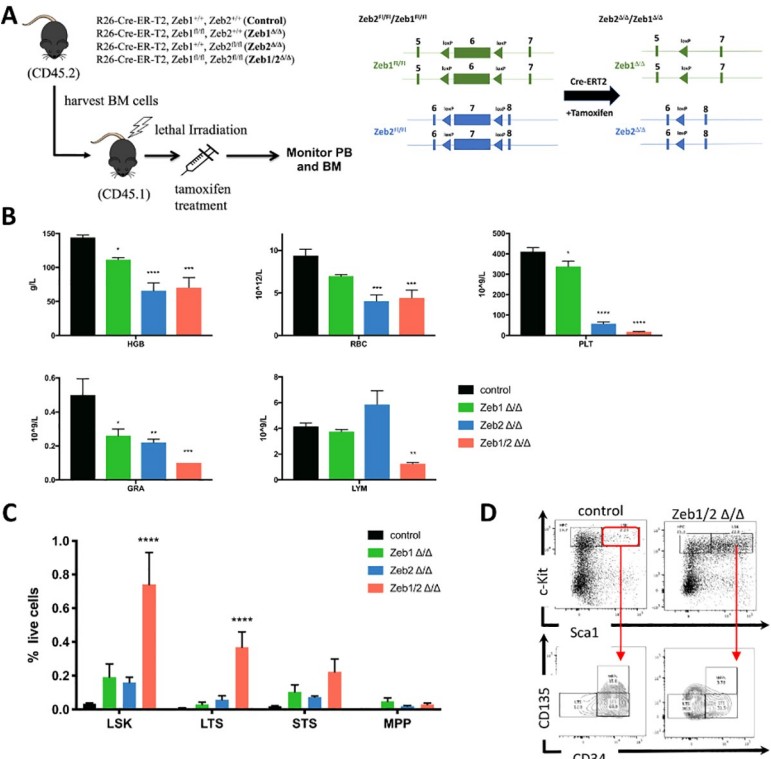

**Fig 4. Double deletion of Zeb1 and Zeb2 causes PB cytopenia and severe differentiation defects in HSPCs. (A)** Schematic of experiments (left panel) used to study the effect of tamoxifen-inducible deletion of *Zeb1*, *Zeb2*, or both after donor BM reconstitution (CD45.2$^+$) of lethally irradiated recipients (CD45.1$^+$). Schematic of *lox*P flanked (floxed-Fl) conditional *Zeb1* and *Zeb2* alleles before tamoxifen induced Cre-mediated deletion and recombined delta *Zeb1* and *Zeb2* alleles after recombination (right panel). **(B)** HCT analysis of BM 10 days after tamoxifen treatment showing decreased HGB (top left) in *Zeb1$^{Δ/Δ}$*, *Zeb2$^{Δ/Δ}$*, and *Zeb1/2$^{Δ/Δ}$* DKO settings. There was as well decreased RBC (top middle), PLT (top right), and GRA (bottom left). LYM (bottom middle) in the BM were decreased only in the Zeb1/2$^{Δ/Δ}$ DKO. **(C)** Flow cytometric analysis of HSPCs in the BM 10 days after tamoxifen treatment showing increased numbers of LSK cells and well as LT-HSCs (LTS- lin$^-$cKit$^+$Sca1$^+$CD34$^-$Cd125$^-$) in *Zeb1/2$^{Δ/Δ}$* DKO settings. **(D)** Representative flow cytometry analysis for hematopoietic stem and progenitor populations showing increased numbers of LSK and LT-HSCs in *Zeb1/2$^{Δ/Δ}$* DKO settings. Bars in panels represent mean ± SD, *n* = 5 per group; *$p < 0.05$; **$p < 0.01$; ****$p < 0.0001$, Dunnett multiple comparisons test. Raw data behind graphs are included in C of S1 Data. BM, bone marrow; DKO, double knockout; GRA, granulocyte; HCT, hematocrit; HGB, hemoglobin; HSPC, hematopoietic stem and progenitor cell; LSK, Lin$^-$Sca1$^+$cKit$^+$; LT-HSC, long-term HSC; LTS, long-term HSC; LYM, lymphocyte; MPP, multipotent progenitor; PB, peripheral blood; PLT, platelet; RBC, red blood cell; STS, short-term HSC.

These results demonstrate that the inducible Rosa26-Cre-ERT2 line can be used in a similar manner as the Mx1-Cre line in the context of BM transfer settings to efficiently examine the cell-autonomous roles of Zeb1/2 in long-term hematopoiesis settings.

To determine the consequences of simultaneous KO of both *Zeb1* and *Zeb2* on steady-state hematopoiesis, similar BM transplants were performed using R26-CreERT2 control, single R26-Cre-ERT2; *Zeb1$^{fl/fl}$* (*Zeb1$^{Δ/Δ}$*), R26-Cre-ERT2; *Zeb2$^{fl/fl}$* (*Zeb2$^{Δ/Δ}$*), and iDKO R26-Cre-ERT2; *Zeb1$^{fl/fl}$*; *Zeb2$^{fl/fl}$* (*Zeb1/2$^{Δ/Δ}$*) BM into lethally irradiated syngeneic recipients. Following recovery from irradiation, recipients were given tamoxifen via oral gavage for 3 consecutive days (**Fig 4A**). Within 2 weeks of the final dose of tamoxifen, the *Zeb1/2$^{Δ/Δ}$* mice had to be euthanized due to anemia, severe weight loss, and general ill health (**S6A Fig**). Immediate early changes in both percentages of live cells and absolute cell numbers of mature blood cells 10 days after tamoxifen showed extreme fluctuations in blood cell composition in the PB in single

*Zeb1*-, *Zeb2*-, and iDKOs compared to R26-Cre-ERT2-only controls (**S6B Fig**: $^*p < 0.05$, $^{**}p < 0.01$, $^{****}p < 0.0001$). BM analysis at this time point showed consistently decreased hemoglobin (HB) levels in all 3 mutant backgrounds, with more severe effects observed in both the *Zeb2*$^{\Delta/\Delta}$ and *Zeb1/2*$^{\Delta/\Delta}$ samples (**Fig 4B**, top left panel). Similar decreases were also observed in red blood cell (RBC) numbers (**Fig 4B**, top middle panel). Platelet (PLT) numbers were also decreased in all 3 mutant backgrounds, but the most severe decreases were observed with *Zeb1/2*$^{\Delta/\Delta}$ samples (**Fig 4B**, top right panel; $^*p < 0.05$, $^{***}p < 0.001$, $^{****}p < 0.0001$). Similar defects in the numbers of granulocytes were observed, again with the most severe decreases in the *Zeb1/2*$^{\Delta/\Delta}$ mice (**Fig 4B**, lower left panel: $^*p < 0.05$, $^{**}p < 0.01$, $^{***}p < 0.001$). Significant decreases in lymphocyte (LYM) numbers were only observed in the *Zeb1/2*$^{\Delta/\Delta}$ samples at this early time of analysis (**Fig 4B**, lower right panel: $^{**}p < 0.01$).

Flow cytometric analysis of HSPC populations from the BM showed a significant increase in the percentage of live LSK and long-term hematopoietic stem cells (LT-HSC-LSK-Cd135$^-$CD34$^-$) per total number of cells in the femur (**Fig 4C and 4D**, **S6C Fig**: $^{****}p < 0.0001$). This is suggestive of a more severe block in multilineage differentiation associated with the inducible loss of both *Zeb1/2* than is observed in either *Zeb2* or *Zeb1* single mutants.

Overall, these results suggest that inducible KO of *Zeb1* and *Zeb2* in the adult hematopoietic system can lead to rapid and lethal decreases in RBC and PLT numbers. These are associated with very early blocks in LT-HSC differentiation that are more severe than genetic inactivation of either *Zeb2* or *Zeb1* alone.

## Continued presence of a single *Zeb2* allele rescues iDKO hematopoietic differentiation

During the establishment of the complex Rosa26-Cre-ERT2; *Zeb1*$^{fl/fl}$; *Zeb2*$^{fl/fl}$ triple transgenic background, we also established cohorts of Rosa26-Cre-ERT2; *Zeb1*$^{fl/fl}$; *Zeb2*$^{fl/+}$ mice with a single wt *Zeb2* allele (*Zeb2*$^{\Delta/+}$), which we included in our subsequent hematopoietic analysis, 10 days after tamoxifen administration. Such *Zeb2*$^{\Delta/+}$ mice did not show any evidence of severe weight loss or anemia. Consistent with this finding, hematocrit (HCT) analysis of mice that carried this allele rescued the HCT defects observed in mice, including restoration of RBC, HB, and PLT cell percentages (**Fig 5A**: $^*p < 0.05$, $^{**}p < 0.01$, $^{****}p < 0.0001$). They also have normalized overall white blood cell (WBC)/LYM and NEU numbers in the PB (**S6D Fig**).

Flow cytometric analysis of the HSPC compartment demonstrated that the presence of a single wt *Zeb2* allele also could normalize the increased percentage of LSK, LT/ST-HSCs present in *Zeb1/2*$^{\Delta/\Delta}$ mice as well as normalize HPC populations, including GMP and MEPs (**Fig 5B and 5C**, **S6E Fig**: $^*p < 0.05$, $^{**}p < 0.01$, $^{****}p < 0.0001$).

Overall, this early analysis of the respective inducible *Zeb1*, *Zeb2* and *Zeb1/2* KOs as well as the presence of a single *Zeb2* wt allele suggests that Zeb2 and Zeb1 can cooperate in controlling hematopoietic lineage fidelity. Further, a single allele of *Zeb2* is enough to normalize these severe defects in hematopoiesis observed in *Zeb1/2* DKOs.

## RNA-seq analysis of LSK-enriched populations reveals both common and unique immediate early gene expression programs controlled by Zeb1 and Zeb2

We performed FACS to isolate LSK cells from the BM of *Zeb1*$^{\Delta/\Delta}$, *Zeb2*$^{\Delta/\Delta}$, and double *Zeb1/2*$^{\Delta/\Delta}$ KO mice, as well as mice carrying a single Zeb2 wt allele (*Zeb1*$^{\Delta/\Delta}$; *Zeb2*$^{\Delta/+}$). LSK cells were harvested 24 hours after the last administered dose of tamoxifen (4 days after tamoxifen treatment was initiated), and RNA was isolated to identify immediate early differentially

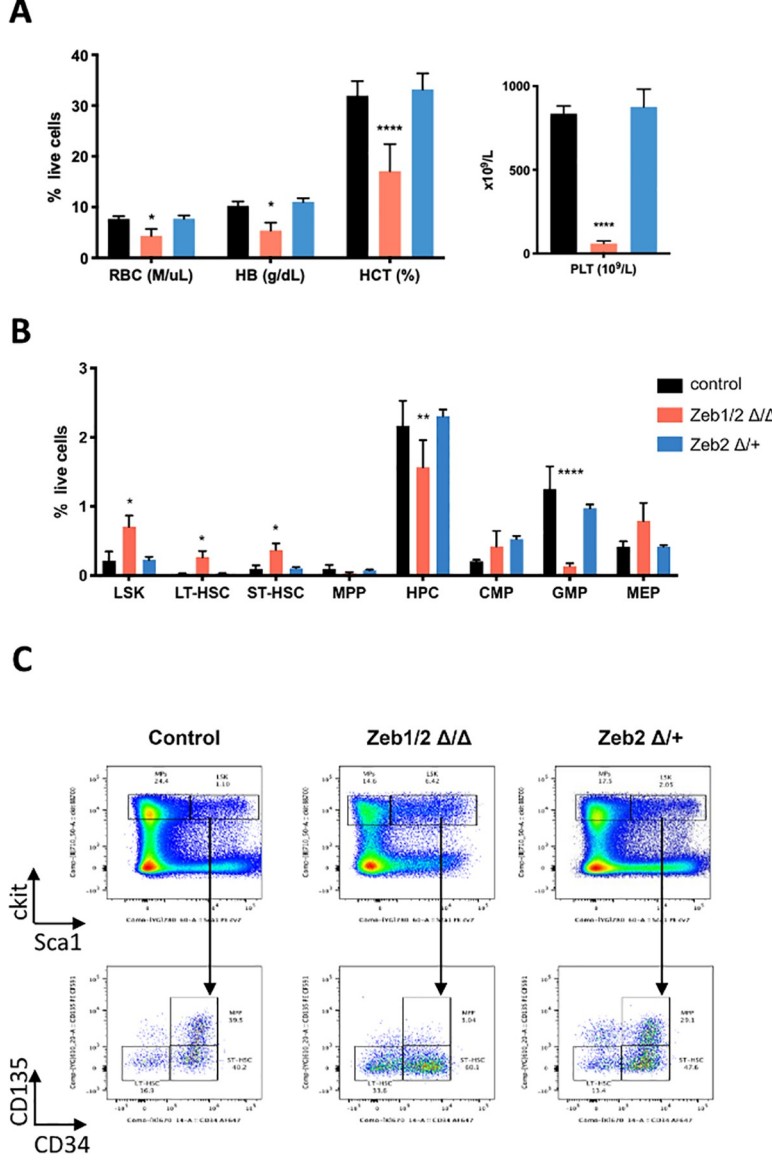

**Fig 5. Maintenance of a single Zeb2 wt allele rescues PB cytopenia and severe differentiation defects in Zeb1/2 DKO HSPCs. (A)** Hemavet analysis showing that maintenance of a single wt *Zeb2* allele in *Zeb2$^{\Delta/+}$*, *Zeb1$\Delta$/$\Delta$* (abbreviated *Zeb2$^{\Delta/+}$*) mice can rescue the HCT defects observed in *Zeb1/2$^{\Delta/\Delta}$* DKO including RBC and HB levels as well as normalization of PLT (right) numbers. **(B)** Flow cytometric analysis of HSPCs from the BM showing that presence of single *Zeb2* allele in *Zeb2$^{\Delta/+}$* mice can also normalize LSK, LT-HSC (lin$^-$cKit$^+$Sca1$^+$CD34$^-$Cd135$^-$), ST-HSC (lin$^-$cKit$^+$Sca1$^+$Cd34$^+$Cd135$^-$), HPC (lin$^-$cKit$^+$Sca1$^-$), CMP, GMP, and MEP HSPC numbers compared to defects observed in *Zeb1/2$^{\Delta/\Delta}$* DKOs. **(C)** Representative flow cytometry plot of HSPCs. Bars in panels represent mean ± SD, *n* = 5 per group; *$p < 0.05$; **$p < 0.01$; ****$p < 0.0001$, Dunnett multiple comparisons test. Raw data behind graphs are included in D of S1 Data. DKO, double knockout; HB, hemoglobin; HCT, hematocrit; HSPC, hematopoietic stem and progenitor cell; PB, peripheral blood; PLT, platelet; RBC, red blood cell; wt, wild-type.

expressed genes (DEGs) between these 4 separate genotypes by RNA-seq (*N* = 3/genotype). This time point was also chosen given the severe lethal phenotype observed in *Zeb1/2$^{\Delta/\Delta}$* mice 10 days post-tamoxifen. Using the RNA-seq data obtained from *Zeb1$^{\Delta/\Delta}$*, *Zeb2$^{\Delta/\Delta}$*, and *Zeb1/2$^{\Delta/\Delta}$* KO LSK cells and bioinformatics analysis, we were able to quantify the degree to which Exon 6 of *Zeb1* and Exon 7 of *Zeb2* were deleted and the residual amount of unrecombined

conditional alleles of *Zeb1* and *Zeb2* that remained at this early time point (**S7 Fig**). Here, we could demonstrate that 4 days of tamoxifen treatment induced approximately 60% loss of Exon 6 in Zeb1$^{\Delta/\Delta}$ *and* Zeb1/2$^{\Delta/\Delta}$ KO cells compared to *Zeb2*$^{\Delta/\Delta}$ LSK cells (**S7A Fig**). Comparing *Zeb2*$^{\Delta/\Delta}$ *and* Zeb1/2$^{\Delta/\Delta}$ KO cells with *Zeb1*$^{\Delta/\Delta}$ LSK cells showed an approximate 90% loss of Exon 7 containing transcripts for *Zeb2* (**S7B Fig**).

Principle component analysis (PCA) of DEGs showed good clustering among samples with the same genotype, with *Zeb1*$^{\Delta/\Delta}$ and single *Zeb2* wt allele samples clustering closer together, whereas *Zeb2*$^{\Delta/\Delta}$ and *Zeb1/2*$^{\Delta/\Delta}$ samples clustered further apart, representing the fact that these are more genetically divergent (**S8A Fig**). Comparing overall DEG heatmaps from *Zeb1*$^{\Delta/\Delta}$ and the 3 other generated genotypes using Degust bioinformatics tools using *Zeb1*$^{\Delta/\Delta}$ samples for the respective comparisons, we found many genes up-regulated in *Zeb2*$^{\Delta/\Delta}$ LSK cells that are further up-regulated in *Zeb1/2*$^{\Delta/\Delta}$ DKOs. A vast majority of these genes are re-repressed, or their normal levels maintained in the single *Zeb2* wt allele "rescue" (**S8B Fig**). These results would imply that Zeb1 and Zeb2 can act together to repress specific gene expression programs involved in hematopoietic differentiation. Volcano plots comparing DEGs between *Zeb1*$^{\Delta/\Delta}$ and *Zeb1/2*$^{\Delta/\Delta}$ samples show the expected decreased expression in *Zeb2* in *Zeb1/2*$^{\Delta/\Delta}$ samples that are null for both *Zeb2* and *Zeb1*, compared with *Zeb1*$^{\Delta/\Delta}$ LSKs that are null for *Zeb1* and are wt for *Zeb2*.

Moreover, further validation of this comparative approach is that some of the top DEGs that are overexpressed in the *Zeb1/2*$^{\Delta/\Delta}$ samples include *Ctse*, *Id2*, and *Epcam* that have been previously identified as being transcriptional targets of Zeb2 that are normally repressed by Zeb2 in T- [29], dendritic [30], and macrophage [31] lineage cells, respectively (**S8C Fig**). These targets among several others are significantly re-repressed or maintained in *Zeb2* single wt allele LSK samples compared with DKO *Zeb1/2*$^{\Delta/\Delta}$ samples (**S8D Fig**). These maintained genes include genes involved hematopoietic adhesion, homing, and niche modulation (including *LSR*, *Ccl6*, *Itga1*, *Ccr9*, *Ccr5*, *Ccr2*, and *Cxcr3*), stemness genes (*Slamf7*, *Lgals3*, *Epcam*, and *Alcam*), transcriptional regulators (*Bcl6*, *Id2*, and *mycl*), and signaling (*Tgfbi*, *Fgfr1*, *Acvrl1*, and *Il13ra1*) (**S8E Fig**).

To identify DEGs relating to immediate early ZEB2-dependent genes more robustly, we have used a bioinformatics approach using a stringent false discovery rate (FDR) <0.1 cutoff [32]. Here, we have compared the DEGs between *Zeb1*$^{\Delta/\Delta}$ and *Zeb1/2*$^{\Delta/\Delta}$ (Zeb2 DEGs), *Zeb1*$^{\Delta/\Delta}$; *Zeb2*$^{\Delta/+}$ versus *Zeb1/2*$^{\Delta/\Delta}$ samples (Zeb2 DEGs rescue), and *Zeb2*$^{\Delta/\Delta}$ versus *Zeb1/2*$^{\Delta/\Delta}$ (Zeb1 DEGs) (**Fig 6A**). These DEGs between these 4 separate genotypes are depicted in a Z-score heatmap (**Fig 6B**). Merging the first 2 datasets, we have identified 143 common DEGs from these 2 Zeb2 genetic settings (total of 321 Zeb2 DEGs, **S2 Table**) (**Fig 6C** left) and 77 Zeb1 DEGs, of which 32 are shared with Zeb2 (**Fig 6C** right, **S2 Table**). Although these numbers of differentially regulated genes are relatively low, they possibly represent some of the first immediate early and possibly direct transcriptional targets of ZEB1/2 that are affected in LSK cells, upon the inducible loss of *Zeb1* or *Zeb2* alone or in *Zeb1/2* DKOs. What is most noticeable again from the heatmap in Fig 6B is a vast majority of DEGs being up-regulated in *Zeb1/2*$^{\Delta/\Delta}$ settings implying a strong degree of cooperation or synergy between ZEB1 and ZEB2 in gene repression (rightmost column in **Fig 6B**). Moreover, there is a subset of DEGs that appear to be up-regulated specifically in *Zeb1*$^{\Delta/\Delta}$ (top leftmost panel in **Fig 6B**), but as well appear to be normalized in *Zeb2*$^{\Delta/+}$ versus *Zeb1/2*$^{\Delta/\Delta}$ samples when one wt *Zeb2* allele is maintained.

Gene ontology (GO) analysis using several sources such as BioCarta, BioCyc, GO, KEGG, and Reactome as queries [33] for ZEB2 differential targets genes identified inappropriate gene activation in LSK cells of genes that should normally only be expressed in more mature immune cells such as those involved in MHC class II antigen presentation, leukocyte migration, or processes reserved T-cell functions including T-cell migration and chemotaxis (**Fig 7A**). Associated network analysis using the STRING database highlighted that many of the

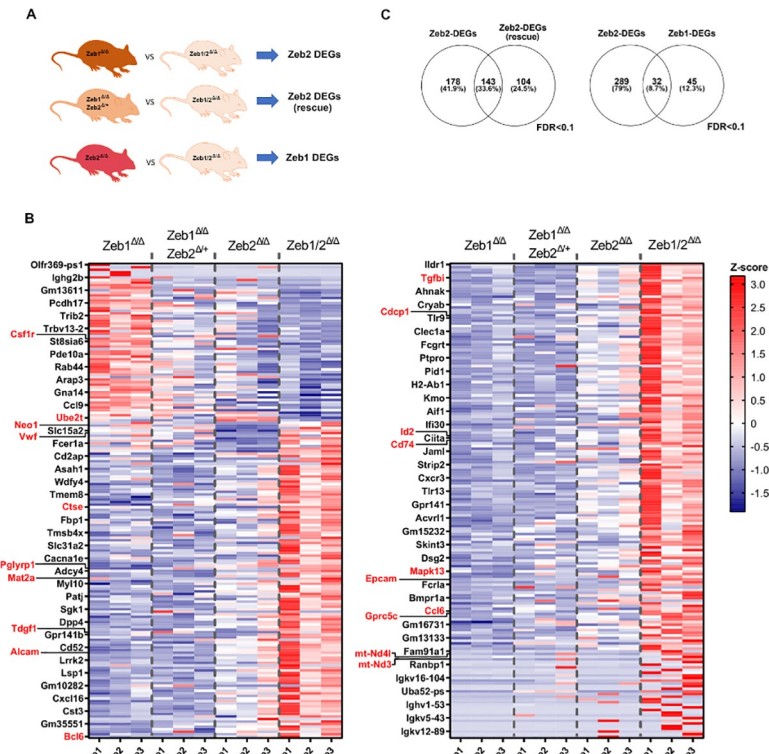

**Fig 6. RNA-seq analysis of LSK-enriched populations reveals both common and unique immediate early gene expression programs controlled by Zeb1 and Zeb2. (A)** DEG lists obtained with edgeRun R package, when comparing R26-Cre-ERT2; *Zeb2*$^{fl/fl}$ (*Zeb2*$^{\Delta/\Delta}$), R26-Cre-ERT2; *Zeb2*$^{fl/+}$, *Zeb1*$^{fl/fl}$ (*Zeb2*$^{\Delta/+}$, *Zeb1*$^{\Delta/\Delta}$), and *Zeb1*$^{fl/fl}$ (*Zeb1*$^{\Delta/\Delta}$) against iDKO R26-Cre-ERT2 *Zeb1/2*$^{fl/fl}$ (*Zeb1/2*$^{\Delta/\Delta}$), respectively. To define DEGs, we used as cutoff an FDR <0.1. **(B)** Heatmap of >360 combined ZEB1 and ZEB2 DEGs, sorted from most induced to repressed ZEB2-DEGs. From left to right, we plotted the Z-scores of gene expression of R26-Cre-ERT2 *Zeb2*$^{fl/fl}$ (Zeb2$^{\Delta/\Delta}$), single R26-Cre-ERT2; *Zeb2*$^{fl/+}$ allele, *Zeb1*$^{fl/fl}$ (*Zeb2*$^{\Delta/+}$, *Zeb1*$^{\Delta/\Delta}$) and the iDKO R26-Cre-ERT2; *Zeb1/2*$^{fl/fl}$ (*Zeb1/2*$^{\Delta/\Delta}$), respectively. **(C)** (Left) Intersections of DEGs from *Zeb1* null cells expressing double or a single *Zeb2* allele against iDKO cells, respectively. (Right) Intersections of DEGs from *Zeb1* or *Zeb2* null cells against iDKO cells, respectively. Raw data behind (B) are included in **S2 Table**. DEG, differentially expressed gene; DKO, double knockout; FDR, false discovery rate; iDKO, inducible double knockout; LSK, Lin⁻Sca1⁺cKit⁺; RNA-seq, RNA sequencing.

DEGs exist in gene regulatory networks controlling chemotaxis as well as antigen processing and presentation via MHC class II that is linked to antimicrobial responses and Toll-like receptor signaling (**Fig 7B**, **S3 Table**). Analysis of intact ZEB1-dependent DEGs shows evidence of aberrant gene regulation involved in mitochondrial functions including respiratory chain complex and NADH dehydrogenase activity (**Fig 7C and 7D**, **S3 Table**).

Using the strict criteria described, only 32 genes were designated as being "shared" between Zeb2-DEGs and Zeb1-DEGs, but this may be significantly underestimated because of the stringent analysis performed and the lack of comparison with Cre and tamoxifen-only controls. Of these 32, several genes (mt-Nd3, mt-Nd4l) involved in mitochondrial metabolism have emerged, as well as other genes such as *Epcam* and *Gprc5c* involved in cancer stemness and dormancy of HSCs, respectively [34,35], and *Cdcp1* that is expressed on leukemic blasts [36]. The common DEG *CCl6* has been implicated in impaired HSC homeostasis [37]. Other interesting Zeb2-only DEGs include *Tgfbi* and *Alcam* that play important roles in HSC biology [38,39] and *Id2* that can significantly affect hematopoietic lineage priming [40]. *Csf1r* is another DEG that plays essential roles in monocyte development [41]. Interestingly, the Zeb1-only DEG Mat2a, an important determinant in MLL-driven AML, is up-regulated in

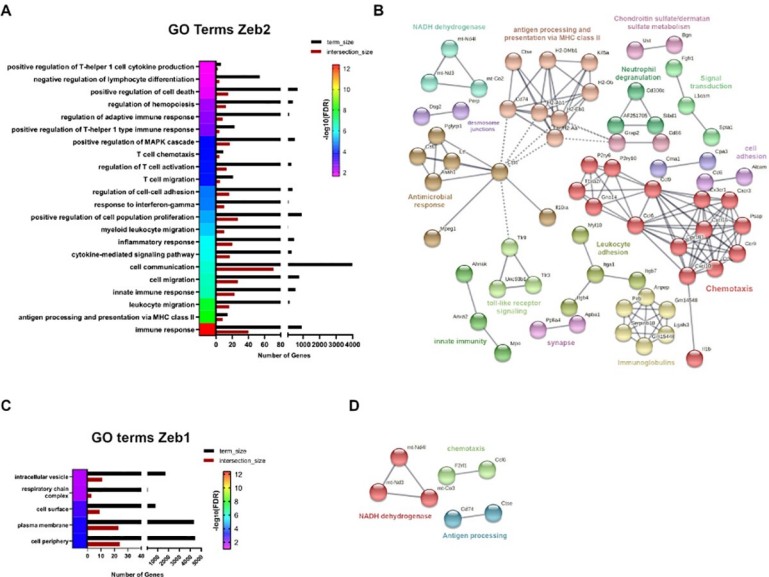

**Fig 7. RNA-seq analysis of LSK-enriched populations reveals both common and unique immediate early gene expression programs controlled by ZEB1 and ZEB2. (A)** Associated GO terms with ZEB2-DEGs, obtained with the STRING database, using an FDR <0.05. **(B)** Associated network analysis for ZEB2-DEGs, obtained with STRING database using highest confidence interaction scores (0.900) and clustered with an MCL inflation parameter of 3. **(C)** Same as (A) for ZEB1 DEGs. **(D)** Same as (B) for Zeb1 DEGs. Raw data behind panels are included in **S3 Table**. DEG, differentially expressed gene; FDR, false discovery rate; GO, gene ontology; LSK, Lin⁻Sca1⁺cKit⁺; RNA-seq, RNA sequencing.

*Zeb1* null settings [42]. Moreover, *Tdgf1* (*Cripto*), *Neo1* (Neogenin-1), and *Mapk13* that play roles in HSC function and stress responses were increased in *Zeb1*-deficient settings [43–45]. Lastly, altered expression of *Vwf* (up) and *Ube2t* (down) genes that are involved in clotting/vascular disorders and Fanconi anemia, respectively, were also DEGs in *Zeb1*^Δ/Δ LSKs [46,47].

To verify if these DEGs identified by RNA-seq can also be directly controlled at the chromatin level by ZEB1 and ZEB2, we reanalyzed publicly available Chromatin Immunoprecipitation Sequencing (ChIP-seq) datasets of ZEB1 and ZEB2 in GM12878 and K562 human cell lines, respectively, plotting the occupancy of these factors in human genes homolog to the DEGs in mouse (see **S4 Table**). Although these 2 cell lines have different hematopoietic origins, we observed marked occupancy of both transcription factors around the gene start, with most of the peaks spanning 0 to 1 kb around these sites (**S9A and S9B Fig**). Moreover, ZEB1 and ZEB2 shared a substantial proportion of peaks and some genes contained ZEB1 and ZEB2 binding enrichment in the same genomic regions (**S9C Fig**). Thus, many genes controlled by Zeb1 and Zeb2 in mouse are also potential transcriptional targets bound by ZEB1/2 in human cells, demonstrating the existence of a conserved network controlled by ZEB1/2.

Overall, we have discovered that many genes may be co-repressed by ZEB1 and ZEB2 that become highly expressed in DKO settings. This finding fits well with the generally accepted view that both ZEB1 and ZEB2 are thought to be predominantly transcriptional repressors in the context of LSK cells and serve to repress genetic programs that should only be activated in more mature lineage-specified cells. Moreover, specific genes were identified that may be associated with altered HSC homeostasis and lineage-specific differentiation/function and are discussed in further detail below.

## Zeb1 and Zeb2 overexpression both drive extramedullary hematopoiesis and monocyte skewing

Previous studies have demonstrated that ZEB1 and ZEB2 may play oncogenic roles in myeloid cell transformation particularly in MLL-AF9 driven models [16,17]. Work presented here would suggest that loss of ZEB1 can interfere with both monocytic and neutrophil differentiation, whereas ZEB2 loss can lead to enhanced granulocytic differentiation in addition to defects (like ZEB1) in monocytic/macrophage differentiation.

In order to examine the effects of ZEB1 and ZEB2 gain of function on hematopoiesis, we have intercrossed conditional Rosa26 locus *Zeb1* [48] and *Zeb2* [15] cDNA-based alleles with the hematopoietic-restricted Vav-iCre/Tie2-Cre lines. Ubiquitous hematopoietic Rosa26 locus-based overexpression (3- to 4-fold as mRNA) of *Zeb1* (**Fig 8B**; $p = 0.0125$) along with EGFP expression (**Fig 8C**) and Zeb2 (**S10A Fig**) in both models led to enlarged spleen sizes already by 3 to 4 months of age that is indicative of extramedullary hematopoiesis (**Fig 8E, S10B Fig**: $^*p < 0.05$). Furthermore, myeloid lineage expansion was observed in both ZEB1 and ZEB2 models in BM, spleen, and PB as judged by increased numbers of CD11b$^+$, Gr1$^+$ cells (**Fig 8F and 8G, S10C and S10D Fig**: $^*p < 0.05$, $^{**}p < 0.01$). Using further Ly6G staining, we confirmed that increased myeloid compartment consisted mainly of CD11b$^+$, Ly6G$^-$ monocytic lineage cells in the BM of both Vav-iCre Rosa26-Zeb1$^{tg/tg}$ and Tie2-Cre driven Rosa26-Zeb2$^{tg/tg}$ models (**Fig 8H**, left and right, respectively; **S10E Fig**: $^*p < 0.05$, $^{**}p < 0.01$).

Overall, these results are consistent with the fact that *Zeb1* and *Zeb2* levels are both essential for monocytic differentiation and can also drive monocytic lineage differentiation when their expression levels increase but are more divergent in their roles in granulocytic lineage differentiation.

Of note, hematopoietic-restricted expression of *Zeb2* does lead to the development of ETP-ALL from 6 months of age onward [15] and only leads to AML development when intercrossed onto a p53 conditional null tumor-prone background [15]. Hematopoietic *Zeb1* overexpressing mice do not spontaneously develop T-cell or myeloid malignancies up to 1.5 years of age and appear to have a normal life expectancy [48].

## Lack of synergy between *Zeb2* and *Zeb1* loss in influencing MLL-AF9 driven leukemic progression

Previous work has determined that ZEB2 is a key genetic determinant in AML initiation/progression and that *Zeb2* knockdown (KD) in retroviral MLL-AF9 models of AML decreases cellular proliferation and enhances myeloid differentiation in vitro [16]. Using novel inducible mouse models of MLL-AF9 myeloid transformation, Zeb1 KD was found to decrease proliferation, increase adhesion, and decrease migratory properties of AML cells in vitro and decreased cell infiltration of AML cells into the BM and other organs in vivo [17]. These results together with their common detrimental effects on myelopoiesis would suggest potential synergy of Zeb1/2 overexpression in AML progression/maintenance.

To examine the role of Zeb2 in AML progression in vivo, we have used Rosa26-CreERT2; *Zeb2*$^{fl/fl}$ BM and an MLL-AF9 retrovirus to create primary AML leukemic cells that allowed us the ability to temporally inactivate *Zeb2* in vivo in a tamoxifen-inducible manner after secondary engraftment (**Fig 9A**). Here, we could demonstrate that inducible *Zeb2* inactivation could significantly enhance the survival of mice transduced with MLL-AF9 secondary leukemic cells compared to Cre-only controls (**Fig 9B**; $p = 0.0018$). In a separate set of experiments tamoxifen-inducible KO of both *Zeb1* and *Zeb2* (although significant) did not extend the survival of mice transplanted with secondary MLL-AF9 AML (**Fig 9C**; $p = 0.0031$) over *Zeb2* inactivation alone and if anything, decreased the overall survival between the 2 groups (70 days for *Zeb2*

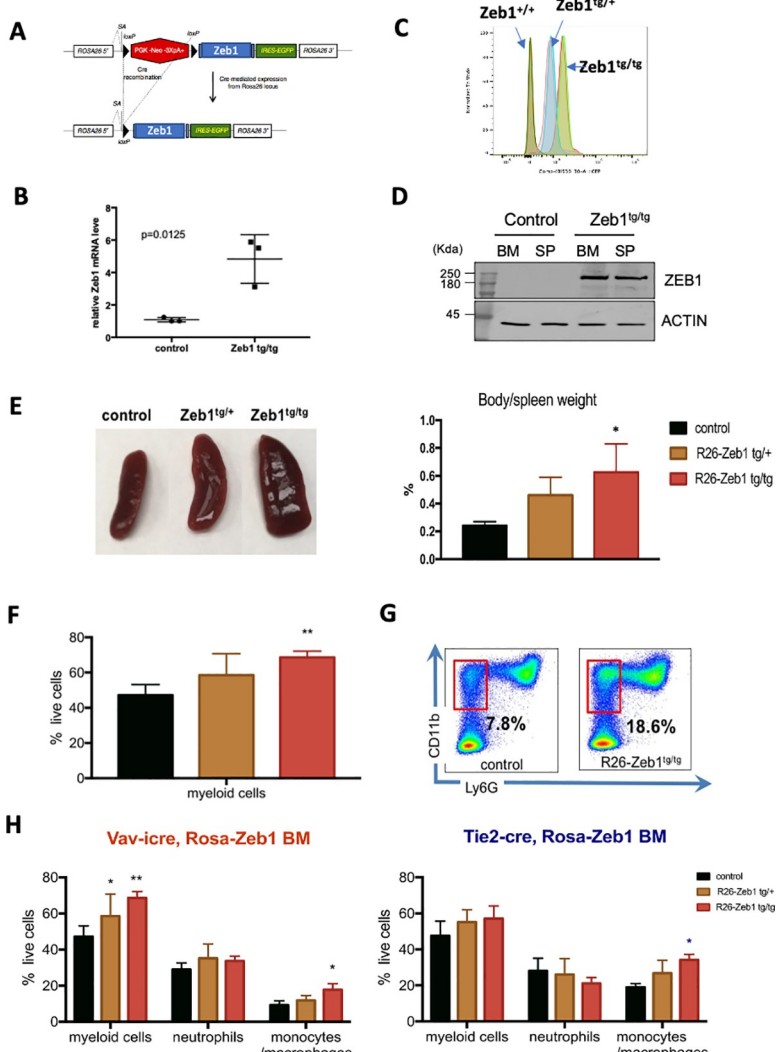

**Fig 8. Zeb1 overexpression leads to extramedullary hematopoiesis/splenomegaly, enhanced myeloid cell development, and monocyte lineage skewing.** (A) Schematic of conditional Rosa26-*Zeb1*-IRES-EGFP-pA + transgenic locus (left). (B) Following Vav-iCre-mediated deletion of the *lox*P flanked transcriptional stop cassette *Zeb1* expression is increased approximately 4.5-fold in transgenic BM HSPCs compared to controls (*N* = three 5-month-old female mice/genotype, *p* = 0.0125; error bars indicate SD of the mean, Mann–Whitney test) along with (C) dosage-dependent EGFP expression in both heterozygous and homozygous *Zeb1* transgenic HSPCs (Flow cytometry for EGFP). (D) Western blot confirmation of increased ZEB1 protein in the BM and spleen of Vav-iCre; *Zeb1*<sup>tg/tg</sup> mice compared to Cre-negative control samples. (E) Increased spleen size/extramedullary hematopoiesis seen in *Zeb1*<sup>tg/tg</sup> transgenic mice (left panel) showing roughly doubling in size compared to body weight (right panel). (F) Flow cytometric analysis showing increased myeloid cells in spleen (CD11b$^+$, Gr1$^+$). (G) Representative flow cytometry plot showing increased CD11b$^+$, Lys6G$^-$ monocytes in the BM of *Zeb1*<sup>tg/tg</sup> mice. (H) Increases in myeloid cells (CD11b$^+$, Gr1$^+$) with monocytic skewing/expansion was present in both Vav-iCre (left) and Tie2-Cre models (right). Data are represented as mean + SD from 3 biological replicates/genotype. $^*p < 0.05$; $^{**}p < 0.01$, nonparametric *t* test. Raw data behind graphs and western blot in (D) are included in E of S1 and S2 Data, respectively. BM, bone marrow; HSPC, hematopoietic stem and progenitor cell; SP, spleen.

KO alone versus 58 days median survival for *Zeb1/2* DKO; **Fig 9B and 9C**). Of note, we documented significant effects of induced Cre activity alone on enhancing survival that is probably related to activation of DNA damage response pathways elicited by Cre binding to pseudo *lox*P sites in the genome [49,50]. Moreover, there were differences in the onset of lethality in

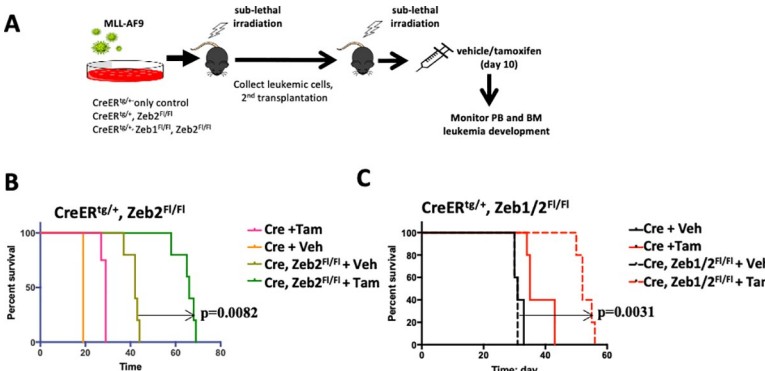

**Fig 9. Inducible deletion of Zeb1/2 increases in vivo survival in MLL-AF9 secondary transplant settings. (A)**
Schematic of inducible deletion strategy to investigate the effects of *Zeb2* and *Zeb1/2* deletion on secondary leukemia
progression. **(B)** Tam-induced *Zeb2* deletion was found to significantly increase overall survival of mice transplanted
with MLL-AF9 secondary tumor cells compared to nontreated Veh treated controls (median survival 66 versus 42
days, *p* = 0.0082, Mantel–Cox test). There was a significant effect of Tam in Cre only treated samples compared to Veh
controls (median survival 29 versus 19 days, *p* = 0.0082, Mantel–Cox test). **(C)** Tam-induced deletion of both *Zeb1* and
*Zeb2* was also found to significantly increase overall survival of mice transplanted with MLL-AF9 secondary tumor
cells compared to nontreated Veh treated controls (median survival 58 versus 31 days, *p* = 0.0031, Mantel–Cox test).
There was a significant effect of Tam in Cre only treated samples compared to Veh controls (median survival 42 versus
31 days, *p* = 0042, Mantel–Cox test). *N* = 5 mice/treatment group for all arms of the experiments. Raw data behind
graphs are included in F of S1 Data. BM, bone marrow; PB, peripheral blood; Tam, tamoxifen; Veh, vehicle.

the Cre-only + tamoxifen controls in the 2 groups (30 days for Cre alone + tamoxifen versus
42 days median survival for Cre alone + tamoxifen in the second set of experiments; **Fig 9B
and 9C**). This may potentially reflect differences in the numbers of tumor-initiating cells present and/or DNA damage response in the individual cell populations between the separate
experiments.

We also reviewed the expression levels of *ZEB1* and *ZEB2* across AML samples available in
cBioPortal database [51]. As evidenced in mouse, the levels of *ZEB1* are in general statistically
lower than *ZEB2* in primary AML cells, with the translocation driver mutation likely affecting
the expression of these transcription factors (**S11A Fig**, left and right, respectively: *p* < 0.05,
**p* < 0.01, ***p* < 0.001, ****p* < 0.0001). Kaplan–Meier survival analysis indicate that overall
higher levels of *ZEB1* are associated with poor prognosis and decreased overall survival irrespective of translocation, supporting an oncogenic role of *ZEB1* (*p* = 0.048 Bonferroni adjusted
*t* test, see **S11B Fig**, left and right, respectively). Conversely, *ZEB2* levels did not appear to statistically influence prognosis in human AML (*p* = 1.0 Bonferroni adjusted *t* test) with regard to
overall survival. Further analysis of *ZEB1* and *ZEB2* mRNA expression demonstrates that the
encoded transcription factors are significantly higher in the leukemic blast population than in
the bulk tumor population (**S11C Fig**: *p* = 0.0417 and *p* = 0.0024, respectively). Therefore,
expression of the mRNAs for these transcription factors may be diluted in the bulk RNA-sample analysis (**S11A Fig**) as well as in overall levels of expression used in the Kaplan–Meier survival curves (**S11B Fig**).

## Discussion

ZEB2 plays essential roles in virtually all aspects of hematopoiesis including regulating embryonic and adult hematopoietic migration and differentiation [8,9]. Using lineage-restricted Cre
lines, ZEB2 has been demonstrated to be essential in regulating T-cell effector and memory
cell state changes during infection [29], controlling macrophage tissue heterogeneity [31], as
well as natural killer (NK) and dendritic cell differentiation and function [30,52]. The role of

the other ZEB family member ZEB1 in hematopoiesis has remained more enigmatic, due predominantly to the neonatal lethality associated with constitutive *Zeb1* KOs [10] and the lack in the field of conditional ("floxed") *Zeb1* alleles. The more recently established conditional loss/gain floxed *Zeb1* mice [24,48] has allowed us as well as others [53] to address the role of ZEB1 in hematopoiesis and hematopoietic transformation, as well as other developmental and disease processes. Using constitutive hematopoietic-restricted Cre lines and hematopoietic transplant settings, we could demonstrate that KO of *Zeb1* during embryonic hematopoiesis does not result in overt embryonic lethality or cephalic hemorrhage that was previously observed for *Zeb2* [8]. However, we did observe slightly decreased Mendelian rates of Cre$^{Tg/+}$; *Zeb1* $^{fl/fl}$ (*Zeb1*$^{\Delta/\Delta}$) mice for both the Tie2-Cre and Vav-iCre lines at weaning. The reason for this submendelian inheritance may imply some embryonic lethality associated with loss of ZEB1 in either endothelial or hematopoietic cells but needs to be further investigated. That may as well reflect differences in the genetic background between this and previous studies or the fact that ZEB2 is dominant of ZEB1 during embryonic hematopoiesis.

Using BM transplants of E14.5 Tie2-Cre; *Zeb1* $^{fl/fl}$ HSPCs into syngeneic lethally irradiated C57BL/6 recipients, we could demonstrate essential roles of ZEB1 in adult hematopoiesis, particularly in monocyte development and in the maintenance of ST-HSC and MPP populations. Using both methylcellulose replating assays and competitive BM reconstitution analysis in long term reconstitution experiments, we could demonstrate that loss of ZEB1 compromises self-renewal and differentiation potential of all hematopoietic lineages. These phenotypes show some similarities, but also some important differences that are associated with loss of ZEB2 in the adult hematopoietic system. *Zeb2*-deficient HSPCs did show similar blocks in monocytic differentiation as *Zeb1* null HSPCs, but also demonstrated enhanced mature granulocytic differentiation defects as well as expanded LSK, LT-HSC, MEP, and GMP progenitors [9]. In both *Zeb1*- and *Zeb2* null BM competitive reconstitution settings, *Zeb1* and *Zeb2* null cells were outcompeted for contribution to the hematopoietic system by wt competitor cells.

This similarity in defects in multilineage differentiation between *Zeb1* and *Zeb2* loss-of-function models prompted us to investigate whether inducible KO of both *Zeb1/2* could lead to more severe defects in hematopoietic differentiation. This is in fact what we observed as *Zeb1/2* DKO mice had to be euthanized due to weight loss and severe defects in hematopoietic differentiation, particularly within the erythroid/megakaryocytic lineages responsible for steady-state RBC and PLT formation. There was as well a significant expansion of LSK and LT-HSCs in the PB and BM that was more severe than the defects seen in either single *Zeb1*$^{\Delta/\Delta}$ or *Zeb2*$^{\Delta/\Delta}$ background. All of these defects were not observed when a single endogenous *Zeb2* allele was present (in Cre+; *Zeb1* $^{fl/fl}$; *Zeb2*$^{fl/+}$ mice). Despite the fact that ZEB2 is a transcription factor that shows haploinsufficiency in Mowat–Wilson Syndrome patients [54], the presence of a single wt allele is sufficient to rescue the *Zeb1/2* DKO defects. Interestingly, these blocks in HSC differentiation appear to occur at the transition point from ST-HSC➔MPP1 (**Fig 1A**) where *Zeb2* mRNA levels undergo transient decreases and *Zeb1* mRNA levels undergo a transient increase (**Fig 1C**). This is reminiscent of Gata1/2 switching that is essential for ensuring erythroid lineage fidelity [55] or the *Zeb1/2* switch described during T-cell development or other hematopoietic lineages [6]. The reasons for and mechanisms involved in this tight control of this Zeb2/1 switch during early hematopoiesis remains to be fully understood.

To gain molecular insight into these findings, we performed RNA-seq experiments on LSK cells within 4 days of initial tamoxifen-induced single *Zeb1*$^{\Delta/\Delta}$ and *Zeb2*$^{\Delta/\Delta}$, and *Zeb1/2*$^{\Delta/\Delta}$ contexts, as well as mice carrying a nonexcised single *Zeb2* wt allele.

Given the technical complexity of this analysis, we did not include Cre-only or tamoxifen-only control samples for comparative purposes given that all samples were Cre$^{+}$ and had received tamoxifen, and therefore had a similar background, with the only changes being loss

 

of *Zeb1* or *Zeb2* or both, or maintaining a single endogenous Zeb2 allele. Previous DEG analysis of Zeb2-deficient LT-HSCs waited 8 weeks after interferon-mediated KO to examine Zeb2 targets [9], and, therefore, this analysis has most probably identified more immediate early changes in gene expression as well as potential direct transcriptional targets of Zeb1/2. Globally, we found that compared to $Zeb1^{\Delta/\Delta}$, the $Zeb2^{\Delta/\Delta}$ LSKs had 321 genes that showed altered gene expression (264 up-regulated and 57 genes down-regulated), and around half of these genes (143) showed normalized expression in LSKs that maintained 1 wt endogenous *Zeb2* allele. GO analysis indicated that many of these genes play roles in mature B-, T-, and myeloid cell functions, suggesting that Zeb2's main job together with Zeb1 is to repress more mature cell lineage programs from being expressed (or to restrain excessive lineage priming) in LSK cells to maintain stem cell pools as well as lineage fidelity. A similar analysis was performed to identify potential Zeb1 targets by examining DEGs between *Zeb2* KO and DKO LSKs, but here a more modest number of genes (77, 66 up and 11 down) were identified. Some of these DEGs play roles in mitochondrial metabolism. That ZEB1 (and potentially ZEB2) may control mitochondrial genes to regulate the survival and metabolism of HSPCs and may play important roles in self-renewal [56] needs to be further investigated.

*Id2* is one of the top differentially regulated genes in *Zeb2* and *Zeb1/2* DKO settings and is an interesting Zeb2 target that is repressed in T-cell subsets and dendritic cells [29,30], and, now, we have also identified *Id2* as a putative direct target that is repressed by Zeb2 (and potentially Zeb1) in LSK cells. This is important in that work previously performed by John Dick's lab has shown that increased ID2 levels in the human hematopoietic system can block lymphoid lineage priming in favor of enhanced HSC numbers as well as enhance priming toward myeloid lineage cells [40]. This is exactly the phenotype observed in the *Zeb2*-deficient hematopoietic system where *Id2* levels are elevated [40]. However, in the double *Zeb1/2* KO system, where Zeb1 is essential for differentiation past the MPP/MPP1 stage, we have dramatic effects on further increasing HSC numbers because the myeloid/lymphoid differentiation pathways become blocked.

Taking a deeper dive into some of these DEGs, several genes including *Epcam1* and *Gprc5c* were found to be up-regulated in both *Zeb1*- and *Zeb2* null settings. EpCAM is an adhesion molecule known to play essential roles in cancer stemness biology of mainly epithelial derived cancers [34]. The role of up-regulated *Epcam* expression in the observed HSPC phenotypes remains to be determined. *Gprc5c* expression has recently been demonstrated to be associated with HSC dormancy [57], although it is unclear what role enhanced it may play on HSPC function. Likewise, *Alcam* and *Tgfbi* levels were found to be increased in *Zeb2*-deficient LSKs. *Alcam* has previously been demonstrated to positively regulate HSC engraftment and self-renewal [38], and its increased expression may simply mirror the increased LT-HSCs numbers present in *Zeb1/2*-deficient mice, which clearly have differentiation blocks. Enhanced *TGFBI* expression has, however, been demonstrated to negatively impact on human HSC differentiation potential at least in in vitro coculture settings [39]. Likewise, enhanced *Ccl6* expression is a common ZEB1/2 DEG whose up-regulated expression impairs HSC homeostasis [37] that could have interfered with the reconstitution potential observed in *Zeb1* and *Zeb* null competitive HSPC transplants.

Up-regulated ZEB1 DEGs that may also have effects on HSCs include *Tdgf1* (*Cripto*), *Neo1*, and *Mapk13*. All 3 of these genes, however, are known to be up-regulated during HSC stress, and their up-regulation may simply reflect the altered BM environment including increased hypoxia caused by the lack of myeloerythroid differentiation [43–45].

To understand the differentiation blocks observed in the monocyte lineage, *Csfr1* appears to be a gene that is positively regulated by ZEB2 that, in its absence, leads to decrease of *Csfr1* expression. *Csf1r* encodes for the colony stimulating factor receptor and is essential for

monocyte/macrophage differentiation [41]. Decreased *Csfr1* expression in *Zeb2*- and potentially *Zeb1* null HSPCs could contribute to the loss of monocyte development observed in *Zeb1* and *Zeb2* individual KO and DKO settings as well as the enhanced monocyte skewing observed in Rosa26 locus based Zeb1/2 transgenic cDNA-based overexpression models as well as potential roles in AML progression [58]. Previously, excessive human recombinant CSF-1 was given to mice and was found to lead to similar splenomegaly and excessive monocyte differentiation phenotypes reported here in this study [59]. Recently, we have demonstrated that in Snai1-induced AML, *Csf1r* is also up-regulated [60]. Whether *Csf1r* is a direct transcriptional target of ZEB1/2 or SNAI1 remains to be determined.

Another intriguing observation is that mice that have deleted *Zeb1/2* become moribund around 2 weeks after tamoxifen-induced KO. This phenotype was characterized by dramatic decreases in PLT and RBC numbers. *Zeb2* null mice have previously been demonstrated to have defects in MEP differentiation into megakaryocytes and erythrocytes [9]. This phenotype was exacerbated by further inactivation of *Zeb1*. Zeb1 DEGs that may have contributed to these defects or other vascular related pathologies include *Vwf* whose expression was increased and *Ube2t* whose expression was decreased in the absence of Zeb1. Increased VWF has been implicated in vascular dysfunction [46] and decreased UBE2T in Fanconi anemia [47], and acute deregulated expression of these genes may have contributed to the adverse outcomes observed within 2 weeks of *Zeb1/2* deletion. Lastly, OSM receptor deficiency was found to lead to severe defects in erythroid and megakaryocytic differentiation [61], and *Osm* transcript levels are decreased in *Zeb1/2* DKO LSK cells.

Clearly, no one gene or pathway alteration can explain every aspect of the complex hematopoietic phenotypes observed in this study. However, we believe that we have identified several unique targets that remain to be further functionally validated concerning their roles in ZEB1/2-mediated control of hematopoietic lineage fidelity. We have summarized our findings of this manuscript as well as others in **Fig 10**.

In terms of the role of Zeb family members in AML progression, we have recently demonstrated that R26 locus and cDNA-based overexpression of SNAI1, which, like ZEB1/2, has also been implicated in EMT processes [13], can drive spontaneous AML formation in mouse settings [60]. In pre-leukemic settings, SNAI1 was found to drive altered myeloid development [60]. Here, we have demonstrated that Rosa26 locus expression of both *Zeb1* and *Zeb2* cDNAs can drive extramedullary hematopoiesis and skew myeloid differentiation toward the monocytic lineage in a similar manner as Snai1. Unlike the situation for *Snai1*, in neither instance, did R26 locus based hematopoietic overexpression of *Zeb1* or *Zeb2* lead to AML development on their own up to 12 months of age. Only on a *p53* null background did such *Zeb2* overexpression led to spontaneous AML development in some rare instances [15]. It is not clear if on similar *p53* null backgrounds ZEB1 may influence myeloid transformation. *Zeb2* overexpression does, however, lead to ETP-ALL development on its own [15], but *Zeb1* overexpression does not [48] as *Zeb1* appears to be a tumor suppressor gene within the T-cell lineage [13].

Previous work has suggested that ZEB1 and ZEB2 [16,17] as well as SNAI1 may play roles in AML transformation, and we have recently shown that *Snai1* deletion can enhance survival in MLL-AF9 as well as AML-ETO/N-RAS AML models in vivo [60]. Here, we show a similar phenomenon for Zeb2 deletion in enhancing survival in MLL-AF9 leukemia. The fact that all 3 EMT transcriptional regulators may be overexpressed in AML and can cause similar disturbances in myeloid development we have begun to ask whether there is synergy between these factors in not only regulating hematopoiesis but in myeloid transformation as well. In this study, we have investigated potential synergy between *Zeb1* and *Zeb2* deletion in modulating AML survival but could not obtain any evidence that the loss of *Zeb1/2* was more significant than *Zeb2* deletion alone in extending survival in MLL-AF9 settings. Previous studies have

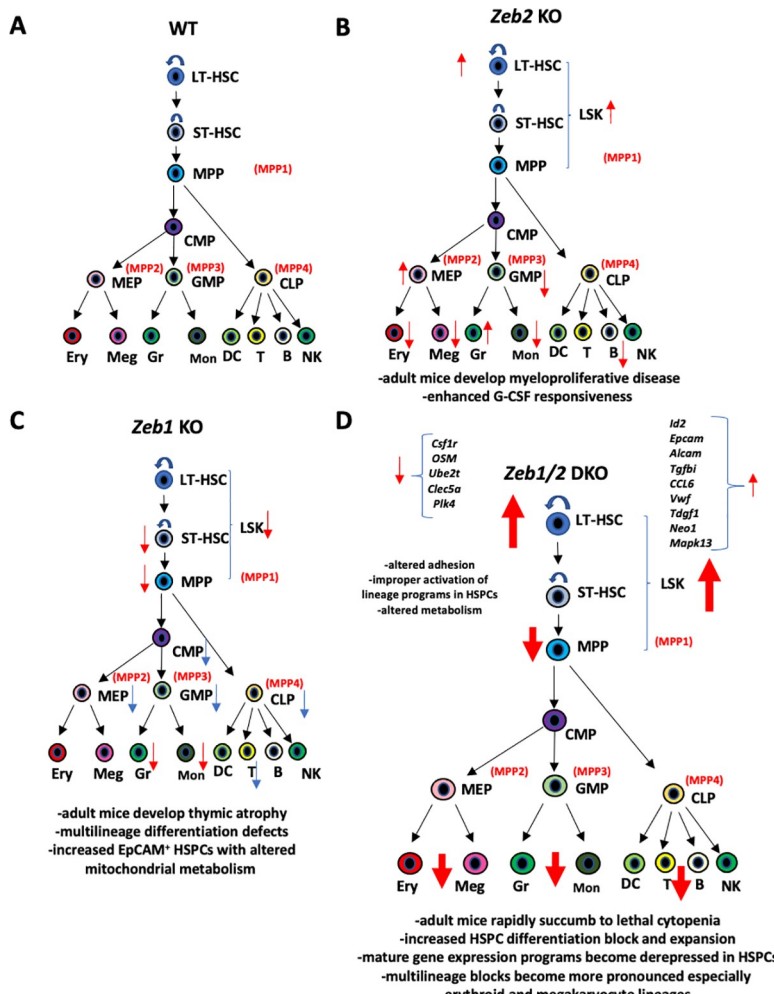

**Fig 10. Summary model of effects of Zeb1, Zeb2, and Zeb1/2 double deletion on hematopoietic system development and steady-state hematopoiesis. (A)** For comparative purposes, a more "classical" view of normal hematopoietic hierarchy is presented with more modern multipotent progenitor nomenclature (MPP1-4) highlighted in red. **(B)** Overview of *Zeb2* null adult hematopoietic phenotypes previously described [9]. Adult mice develop myeloproliferative disease over time that is driven by enhanced G-CSF responsiveness [9] as well as mild differentiation defects in multiple HSPC populations including increased LT-HSCs, increased MEPs, decreased GMPs as well as defects in mature hematopoietic populations including decreased RBC, megakaryocytes, monocytes, and B cells but expanded terminal granulocyte differentiation (red arrows). **(C)** Represents a hybrid summary view between results of this study (red arrows) along with those found by [63] (blue arrows). Unlike *Zeb2* KOs LSK, ST-HSC and MPP numbers are down in Zeb1 hematopoietic null mice and display multilineage differentiation defects with decreased numbers of progenitors and mature hematopoietic cells particularly T cells with mice developing thymic atrophy [62]. HSPC phenotype is characterized by increased EpCAM expression with altered survival and metabolism profiles. **(D)** Inducible loss of both *Zeb1* and *Zeb2* leads to acute BM failure with mice succumbing to lethal cytopenia within 2 weeks. Block in LT-HSC differentiation observed in *Zeb2* KO is exacerbated in *Zeb1/2* DKO settings and multilineage blocks especially in erythroid and megakaryocyte lineages are more severe (increased size of red arrows). Molecular analysis of LSK progenitors highlighting altered migratory and metabolic pathways as well as improper activation of multiple lineage-specific programs normally only observed in mature myeloid and lymphoid cell types. Specific relevant genes that are up- or down-regulated are indicated and further elaborated on in the discussion. BM, bone marrow; DKO, double knockout; G-CSF, granulocyte colony-stimulating factor; HSPC, hematopoietic stem and progenitor cell; KO, knockout; LSK, Lin⁻Sca1⁺cKit⁺; LT-HSC, long-term HSC; NK, natural killer; RBC, red blood cell; ST-HSC, short-term HSC; wt, wild-type.

demonstrated that some degree of myeloid differentiation is required for leukemic transformation [62], and, therefore, given the severe block in differentiation exhibited in *Zeb1/2* KO settings, it will be interesting to determine the effects of *Zeb1* and *Zeb2* deletion on primary AML disease formation. Further studies are clearly required to determine the degree of synergy and crosstalk between ZEB1/2 and SNAI1 in AML as well as their common and unique underlying molecular roles in driving myeloid leukemia.

## Note added in proof

During the final preparation of this manuscript, Almotiri and colleagues published a similar study [63] as ours using the same conditional *Zeb1* allele that we have used in this study [24]. However, their analysis of ZEB1 predominantly used the inducible Mx1-Cre mouse model to delete *Zeb1* alone, whereas this study predominantly used constitutive hematopoietic enhanced Tie2 and restricted Vav-iCre lines with BM transplants to determine the cell-autonomous role of Zeb1 in hematopoiesis. In general, there is a significant agreement between these 2 studies especially pertaining to the monocyte defects and loss of reconstitution potential of *Zeb1*-deficient HSPCs. The focus of their manuscript is more on T-cell defects that were previously reported in the full *Zeb1* null study [11], whereas we focused more on the novel synergistic roles between ZEB1 and ZEB2 in maintaining hematopoietic lineage fidelity. Subtle differences in the results between the 2 papers may originate from the other groups using the Mx1-Cre model, nonhematopoietic expression of Cre in the stroma, as well as potential effects of poly:IC and its known interferon response and potential effects of HSC quiescence [64]. Fetal liver derived HSCs used in our study are more highly proliferative than BM-derived adult HSCs and appear to utilize oxidative metabolic pathways more than BM HSCs and may be better protected from reactive oxygen species (ROS)-mediated genotoxicity [65]. Moreover, human fetal and adult HSCs have been demonstrated to give to distinct T-cell lineages in humans [66]. Given that we do see differences in T-cell differentiation of *Zeb1*-deficient HSPCs in our study compared to Almotiri and colleagues, this may be a contributing factor.

One area where there is some discrepancy between the 2 studies is that Almotiri and colleagues suggest that Zeb1 plays a tumor suppressor role in AML progression whose loss enhances AML lethality. In the present study, we have investigated potential synergy between Zeb1 and Zeb2 loss, and, although loss of Zeb1 did decrease overall survival slightly in *Zeb1/2* DKO MLL-AF-9 settings compared to Zeb2 loss alone, there was still significant survival advantage observed in *Zeb1/2* DKOs compared to Cre-only controls. There was as well significant variability observed in these studies with clear effects of Cre-alone, affecting the outcomes.

Our work with Rosa26-based transgenic overexpression models would suggest equal detrimental effects of *Zeb1* overexpression as *Zeb2* on myeloid differentiation, and our work would tend to support previous reports that Zeb1 can act as an oncogene in certain forms of AML [11]. Additionally, increased levels of *ZEB1* mRNA are associated with decreased overall survival of AML patients that would be consistent with an oncogenic role. It is clear from our work and that of others that the concept of oncogene and tumor suppressor as it pertains to ZEB1 and ZEB2 is very contextual and depends on cell type examined and stage of cell differentiation when *Zeb1/2* mRNA levels become dysregulated [17]. Moreover, from our analysis and the work from Stavropoulou and colleagues [17] elevated expression of *Zeb1* and potentially *Zeb2* transcripts in the leukemic stem cell compartment may ultimately drive poor outcomes as opposed to overall expression of *Zeb1/2* in bulk AML samples.

## Materials and methods

### Animal experimentation and handling

The Tie2-Cre [25], tamoxifen (Tam)-inducible ROSA26$^{CreERT2/+}$ mice [27], and conditional Zeb2-KO [28], Zeb1-KO [24] mouse model have been backcrossed to a C57Bl/6 genetic background for at least 10 generations. All strains were bred in-house in specific pathogen-free (SPF) facility.

E14.5 cells from an entire fetal liver from Tie2-Cre Zeb1$^{+/+}$, Tie2-Cre Zeb1 $^{fl/fl}$ CD45.2$^+$ backgrounds were transplanted via tail vein injection into lethally irradiated ($2 \times 550$ Rads) CD45.1$^+$ hosts (1 fetal liver/host).

Competitive BM experiments were performed using equal numbers ($2.5 \times 10^6$) of CD45.2$^+$ Vav-iCre, Zeb1 $^{fl/fl}$, Vav-iCre Zeb1$^{fl/+}$, or Vav-iCre only doner cells and CD45.1+ competitor cells that were transplanted by tail vein injection into lethally ($2 \times 550$ Rads) irradiated CD45.1 hosts.

For inducible deletion experiments, $5.0 \times 10^6$ BM donor cells were used from CD45.2$^+$ ROSA26$^{CreERT2/+}$ Zeb1$^{+/+}$ Zeb2$^{+/+}$, ROSA26$^{CreERT2/+}$ Zeb2$^{fl/fl}$, ROSA26$^{CreERT2/+}$ Zeb1$^{fl/fl}$, ROSA26$^{CreERT2/+}$ Zeb2$^{fl/+}$ Zeb1$^{fl/fl}$, and ROSA26$^{CreERT2/+}$ Zeb1/2$^{fl/fl}$ backgrounds to reconstitute lethally irradiated ($2 \times 550$ Rads) CD45.1 recipients. These mice were allowed to recover for between 2 and 6 weeks before they were orally gavaged for 3 consecutive days with tamoxifen (5 mg in 50-ul vehicle per 25-g body weight).

In MLL-AF9 initiation and progression experiments, MLL-AF9 transduced C57BL/6 (CD45.2) fetal liver cells (ROSA26$^{CreERT2/+}$, Zeb1$^{+/+}$, Zeb2$^{+/+}$, ROSA26$^{CreERT2/+}$ Zeb2$^{fl/fl}$ or ROSA26$^{CreERT2/+}$, Zeb1$^{fl/fl}$, Zeb2$^{fl/fl}$) were collected under aseptic conditions and were intravenously injected into lethally irradiated ($2 \times 550$ Rads) C57BL/6 (CD45.1) recipient mice. For secondary leukemia transplantation studies, 10,000 GFP$^+$ckit$^{hi}$CD11b$^+$ primary leukemia cells were intravenously injected into sublethally irradiated (550 Rad) recipient mice. Inducible deletion of the *Zeb* gene was achieved following exposure of CreERT2, Zeb1$^{fl/fl}$, Zeb2$^{fl/fl}$ mice to tamoxifen (5-mg tamoxifen in 50-ul vehicle per 25 g) by oral gavage once daily for 3 consecutive days. All irradiate mice were maintained on acidified water following irradiation.

All experiments were performed according to the regulations and guidelines of the Ethics Committee for care and use of laboratory animals of Monash University (E/1690/2016/M, #5789) and the University of Manitoba (#18–050).

### Flow cytometric and HCT analysis

Cells were stained with antibodies listed in B in **S5 Table** according to the manufacturer guidelines. Flow cytometric analyses were performed on the LSRII and Fortessa X-20 cytometer (BD Biosciences, Sydney, Australia), and the results were analyzed by FACSDiva or FlowJo software (BD Biosciences). Cells for MLL-AF9 experiments and RNA-seq were stained and sorted on Influx or FACSAria Fusion sorters (BD Biosciences) at AMREP Flow Cytometry Core Facility and FlowCore, Monash University. All flow cytometry data generated and/or analyzed during the current study are available on Zenodo (doi: 10.5281/zenodo.5498282).

Submandibular blood samples were collected into EDTA-coated tubes, and hematology parameters were measured using a HemaVet 950FS automated blood analysis machine (Drew Scientific, Miami Lakes, Florida, USA).

### Methylcellulose culture and replating

E14.5 fetal livers were isolated and cultured in methylcellulose 3434 (Stem Cell Technologies, Vancouver, British Columbia, Canada) for 7 days at 37˚C and 5% $CO_2$ in duplicate. For replating assays, cells were collected and resuspended in 1X PBS and then replated in Methocult 3434 and cultured for another 14 to 21 days.

## RNA-seq analysis

Mouse BM cells were stained using antibodies listed in B of **S5 Table** and LSK cells were sorted into TRIzol Reagent (Thermo Fisher Scientific, Australia). RNA was extracted using Direct-zol RNA microprep kit (Zymo Research, distributed by Integrated Sciences, Chatswood, NSW, Australia) and quality assessed using a BioAnalyzer machine (Agilent Mulgrave, Australia). Library preparation was performed using the Nugen RNA-Seq system V2 (SPIA Amplification) followed by Nugen Ultralow Library, and single-end 75 bp reads were generated on an Illumina NextSeq 500 machine in high output settings. Details of RNA-seq data analysis and exon quantification are included in S1 Text. Raw RNA-seq data files were deposited as a NCBI BIOPROJECT #PRJNA679880.

scRNA-seq data of HSPCs previously published [19] was reanalyzed using pseudotime and trajectory-based analysis of scRNA-seq data [20,21] to gain insight into the levels and temporal expression of Zeb1 and Zeb2 during early hematopoiesis. Details of this analysis are included in S1 Text.

## Real-time quantitative PCRs

Total RNA was isolated using RNeasy Plus Mini Kit (Qiagen, Chadstone Centre, Victoria, Australia). cDNA was synthesized using the SuperScript IV First-Strand Synthesis System (Thermo Fisher Scientific, Australia), starting from equal amounts of RNA. qRT-PCRs were performed using the TaqMan Gene Expression Master Mix (Applied Biosystems) on a Licht-Cycler 480 system (Roche, Australia). Gene expression was standardized against housekeeping genes Gusb, Gapdh, and Hprt. All primers used are listed in part A of **S5 Table**.

## Western blot

Mononuclear cells were purified from the BM and the spleen. Whole cell lysates were prepared by using the RIPA Buffer (10mM Tris-HCl, pH 8.0, 1mM EDTA, 0.5mM EGTA, 1% Triton X-100, 0.1% Sodium Deoxycholate, 0.1% SDS, 140mM NaCl dilute with $dH_2O$). The proteins were extracted and run by using the 10% gel (Acrylamide/Bis-acrylamide 29:1). The primary antibodies were Rabbit Ab anti-ZEB2 (NBP1-82991), Rabbit ab anti-ZEB1 (NBP1-05987), and Rabbit ab anti-β-actin coupled to HRP(NB600-503SS). Abs were obtained from Novus Biologicals (Victoria, Australia).

## Statistical analysis

Data were presented as mean ± SD and indicated in the figures. Comparison between 2 data groups was done by 2-sided Student *t* test. Dunnett multiple comparisons test was used for statistical analysis between 3 or more experiment groups. Kaplan–Meier survival curves for MLL-AF9 models were generated with GraphPad Prism 7 software and log-rank (Mantel–Cox) test was performed for statistical analysis.

## Supporting information

**S1 Fig. ImmGen expression data for Zeb1 and Zeb2.** ImmGen normalized RNA expression data from adult mouse hematopoietic system for *Zeb1* and *Zeb2* in various hematopoietic (sub) lineages (top) and lineage hierarchy highlighting common/differential expression between *Zeb1* and *Zeb2* (bottom). Here, increased relative expression is highlighted as red and low expression is indicated in blue. Data generated using online tools at https://www.immgen.org/.
(TIFF)

**S2 Fig. Tie2 and Vav-iCre mediated deletion of Zeb1 leads to sub-mendelian rates of transmission at weaning but no signs of embryonic lethality. (A)** Phenotypically normal control and Tie2-Cre; *Zeb1*$^{fl/fl}$ embryo at E14.5. **(B)** Table of expected and observed Tie2 and Vav-iCre; *Zeb1*$^{fl/fl}$ mice at P21. E14.5, embryonic day 14.5; P21, postnatal day 21.
(TIFF)

**S3 Fig. Hematopoietic-specific loss of Zeb1 leads to differentiation defects in myeloid lineages and specific HSPC populations. (A)** SLAM marker expression showing similar numbers of LT-HSCs (CD150$^+$CD48$^-$) in Tie2-Cre, *Zeb1* null and control Cre$^-$ reconstituted BM. **(B)** Flow cytometric analysis of HSPC populations within the BM of *Zeb1*-deficient mice identified significant decreases ($^*p < 0.05$) in overall LSK (lin$^-$cKit$^+$Sca1$^+$) numbers but no significant decrease in the total number of stem cells; LT-HSCs (lin$^-$cKit$^+$Sca1$^+$CD34$^-$Cd125$^-$), ST-HSCs (lin$^-$cKit$^+$Sca1$^+$Cd34$^+$Cd135$^-$), and MPPs (lin$^-$cKit$^+$Sca1$^+$Cd34$^+$Cd135$^+$). MPP were analyzed by FcgammaR, CD34 expression to further define MEP, GMP, and CMP populations. A significant decrease in total number of GMPs but no significant changes were observed in total numbers of CMP or MEPs in *Zeb1*-deficient BM compared to controls. **(C)** Flow cytometric analysis of PB of reconstituted mice showed defects in *Zeb1* null HSPC contribution to myeloid cells (Cd11b$^+$) including monocytic (Cd11b$^+$Ly6G$^-$) and NEU (Cd11b$^+$Ly6G$^+$) lineage cells. Here, absolute cell number/femur is given. **(D)** Representative cytometry plot of data shown in (C). **(E)** B220 B cell marker analysis showing no significant differences in % or total B cells in *Zeb1* null and control reconstituted BM. Here, absolute cell number/femur is given. Error bars indicate SD of the mean ($n = 4$ per group, $^*p < 0.05$). Raw data behind graphs are included in A of S1 Data. BM, bone marrow; HSPC, hematopoietic stem and progenitor cell; LSK, Lin$^-$Sca1$^+$cKit$^+$; LT-HSC, long-term HSC; MPP, multipotent progenitor; PB, peripheral blood; ST-HSC, short-term HSC.
(TIFF)

**S4 Fig. Hematopoietic changes associated with tamoxifen-inducible loss of Zeb2 in BM recipients reconstituted with R26-CreERT2; Zeb2$^{fl/fl}$ BM. (A)** HCT analysis showing decreased WBC, HGB, and PLT numbers as well as **(B)** decreased numbers of granulocytes, monocytes, and B cells but increases in mature granulocytes associate with tamoxifen-inducible deletion of *Zeb2* in the adult BM. These phenotypes were previously observed in interferon induced Mx1-Cre mediated deletion of Zeb2 [17]. Raw data behind graphs are included in G of S1 Data. BM, bone marrow; HCT, hematocrit; HGB, hemoglobin; PLT, platelet; WBC, white blood cell.
(TIFF)

**S5 Fig. Tamoxifen-induced Cre-mediated hematopoietic loss of *Zeb1* leads to decreased hematopoietic colony formation potential in adult BM as well as altered T-cell differentiation. (A)** Schematic of BM reconstitution experiments and analysis preformed after Cre-mediated deletion of *Zeb1*. **(B)** PCR gel analysis of genomic DNA from total BM of tamoxifen-driven excision of wt (Cre only) or Rosa26-*Zeb1*$^{fl/fl}$ mice. The upper band corresponds to deletion of *Zeb1* loci (Δ, approximately 380 bp), middle band indicates the amplification of flox/flox allele (approximately 300 bp), and lower band corresponds to the amplification of the wt *Zeb1* locus (Cre only, approximately 220 bp). **(C)** HSPCs isolated from *Zeb1*$^{Δ/Δ}$ BMs show decreased numbers of colonies in methylcellulose-based colony assays at the first plating that further decreases at the first (replate1) and secondary replating (replate2) compared to Cre negative controls. Data are represented as mean + SD from 2 biological replicates per condition, each one consisting in 2 technical replicates. $^*p < 0.05$; $^{**}p < 0.01$, nonparametric *t* test. **(D)** Representative flow cytometric analysis of thymus showing overall decreases in thymic

cellularity and **(E)** significantly decreased DN4 (CD25⁻CD44⁻) progenitors and aberrantly expanded CD4⁺ T cells. $N = 3$/genotype *$p < 0.05$; **$p < 0.01$. Raw data behind graphs are included in H of S1 Data. BM, bone marrow; HSPC, hematopoietic stem and progenitor cell; wt, wild-type.
(TIFF)

**S6 Fig. Double deletion of Zeb1 and Zeb2 causes PB cytopenia and severe differentiation defects in HSPCs. (A)** Weight changes over time associated with tamoxifen mediated deletion of *Zeb1*, *Zeb2*, *Zeb1/2*, or Cre negative control mice. **(B)** Flow cytometric analysis showing fluctuating changes in hematopoietic system associated with tamoxifen mediated deletion of *Zeb1*, *Zeb2*, or both *Zeb1* and *Zeb2* 10 days after the last dose of tamoxifen. Data presented are as a percentage of live cells. **(C)** Flow cytometric analysis of HSPCs in the BM 10 days after tamoxifen treatment showing alterations in total cell number/femur following deletion of *Zeb1*, *Zeb2*, or *Zeb1* and *Zeb2*. **(D)** Normalization of HCT and **(E)** HSPC populations associated with the maintenance of a single *Zeb2* allele (blue bars) compared to *Zeb1/2^Δ/Δ* DKOs. HSPC results given are in cell number/femur. Bars in panels represent mean ± SD, $n = 5$ per group; *$p < 0.05$; **$p < 0.01$; ****$p < 0.0001$, Dunnett multiple comparisons test. Raw data behind graphs are included in I of S1 Data. BM, bone marrow; DKO, double knockout; HCT, hematocrit; HSPC, hematopoietic stem and progenitor cell; PB, peripheral blood.
(TIFF)

**S7 Fig. Exome quantification of RNA-seq data showing degree of recombination of conditional Zeb1 and Zeb2 transcripts in Zeb1^Δ/Δ, Zeb2^Δ/Δ and DKO- Zeb1^Δ/Δ; Zeb2^Δ/Δ LSK + cells. (A)** (Left) IGV snapshot of 1× normalized BigWig tracks derived from alignments of *Zeb1^Δ/Δ*, *Zeb2^Δ/Δ* and DKO (*Zeb1^Δ/Δ*; *Zeb2^Δ/Δ*) LSK+ cells RNA-seq in the **Zeb1** locus. From bottom to top, Zeb1^Δ/Δ tracks are colored in green, Zeb2^Δ/Δ tracks are colored in blue, and DKO tracks are colored in red. Genomic scale is expressed as kilobases. Exon numbers are indicated in bold letters, and the arrow indicated the tamoxifen-mediated excised floxed exon. (Right) Normalized counts per genotype of **exon 6** of **Zeb1**. Error bars indicate SD of the mean ($n = 3$ per group, **$p < 0.01$, ***$p < 0.001$, ns = nonsignificant, nonparametric $t$ test). **(B)** (Left) Same as (A, left) around the **Zeb2** locus. Exon numbers are indicated in bold letters, and the arrow indicates the tamoxifen-mediated excised floxed exon. Genomic scale is expressed as kilobases. (Right) Same as (A, right) for the **exon 7** of **Zeb2**. Error bars indicate SD of the mean ($n = 3$ per group, ****$p < 0.0001$, ns = nonsignificant, nonparametric $t$ test). Raw data behind graphs are included in **S6 Table**. DKO, double knockout; LSK, Lin⁻Sca1⁺cKit⁺; RNA-seq, RNA sequencing.
(TIFF)

**S8 Fig. Degust analysis of RNA-seq data of LSK cells obtained from Zeb1, Zeb2, Zeb1/2 DKO, or single Zeb2 allele rescue. (A)** PCA of DEGs showing clustering of samples based upon genotype. *Zeb1^Δ/Δ Zeb2^+/+* samples (yellow) cluster more closely together with *Zeb1^Δ/Δ Zeb2^Δ/+* samples (blue). *Zeb1^Δ/Δ Zeb2^Δ/Δ* DKO (red) and *Zeb1^+/+Zeb2^Δ/Δ* (green) samples cluster father apart because they are more genetically diverse. **(B)** Gene expression heatmap showing that *Zeb1/2* DKO samples have many genes that become up-regulated compared to either *Zeb1* or *Zeb2* single deletion alone and these genes appear to be normalized if even a single *Zeb2* allele is present in *Zeb1^Δ/Δ Zeb2^Δ/+* samples. **(C)** Volcano plot of DEGs highlighting differences in gene expression between *Zeb1^Δ/Δ Zeb2^+/+* and *Zeb1^Δ/Δ Zeb2^Δ/Δ* DKO samples. Highlighted are 3 genes known to be repressed by *Zeb2* including *Ctse*, *Id2*, and *Epcam*. Moreover, Zeb2 is highlighted as being down-regulated between the 2 samples. **(D)** Volcano plot showing how maintained presence of single Zeb2 allele in Zeb1^Δ/Δ Zeb2^Δ/+ samples can lead to

repression or normalization of up-regulated genes observed in *Zeb1/2* DKO samples. FDR rates for (C) and (D) was 0.05. **(E)** Highlighted *Zeb2* DEGs and rough grouping of biological processes associated with each gene. Raw data behind plots are included in **S2 Table**. DEG, differentially expressed gene; DKO, double knockout; FDR, false discovery rate; LSK, Lin⁻Sca1⁺cKit⁺; PCA, principle component analysis.
(TIFF)

**S9 Fig. ZEB1 and ZEB2 chromatin occupancy in human cell lines reveals robust consistency within discovered mouse Zeb1/2 DEGs. (A)** Plot of the average occupancy and the occupancy profiling of ZEB1 and ZEB2 in GM12878 and K562 cell lines, respectively, in 2 different ChIP-seq experiments. Each row represents a human homolog matching mouse DEGs found in LSK cells. The plots are centered at the first ATG of every gene, and rows are sorted by occupancy values from the smallest (top) to highest (bottom) in a 4-kb window. ChIP-seq data were aligned against hg38 human genome, and BigWig files were obtained from each BAM file by using deeptools bamCoverage tool. Each BigWig file was normalized to 1× method using the mappable human genome size. The colormap used in the heatmaps was jet and the missing data color in the plots was dark blue. **(B)** Distribution of transcription factor binding loci relative to the TSS of the plotted genes in (A) upstream and downstream from the TSS of genes, expressed as percentages, obtained with the ChIPseeker R package. Distance to the TSS was plotted with distinctive colors. **(C)** Human gene models and BigWig tracks from the top-enriched gene LSR including MYCL, ITGB7, and SLAMF7 genes presenting ZEB1 and/or ZEB2 peaks in all ChIP-seq datasets (including technical replicates). Raw data behind plots are included in J of **S4 Table.** ChIP-seq, Chromatin Immunoprecipitation Sequencing; DEG, differentially expressed gene; LSK, Lin⁻Sca1⁺cKit⁺; TSS, transcriptionally start site.
(TIFF)

**S10 Fig. Rosa26 locus–based Zeb2 hematopoietic-restricted overexpression leads to extramedullary hematopoiesis/splenomegaly, enhanced myeloid cell development, and monocyte lineage skewing. (A)** Increased spleen size/extramedullary hematopoiesis seen in Tie2-Cre; *Zeb2^{Tg/Tg}* transgenic mice (left panel) showing roughly doubling in size compared to body weight (right panel). **(B)** Flow cytometric analysis showing increased myeloid cells (CD11b⁺, Gr1^{lo}) in the BM (left) and spleen (right). **(C)** Representative flow cytometry plot showing increased CD11b⁺, Gr1^{lo} myeloid cells in the BM of Tie2-Cre; *Zeb2^{Tg/Tg}* mice. **(D)** Summary graph of increased myeloid cells (CD11b⁺, Gr1^{lo}) in heterozygous Tie2-Cre; R26-*Zeb2^{Tg/+}* and homozygous R26-*Zeb2^{Tg/Tg}* BM cells as well as increased monocytes (CD11b⁺, Lys6G⁻) present in R26-*Zeb^{Tg/Tg}* BM cells. **(E)** Western blot analysis showing increased ZEB2 protein expression in Vav-iCre; *Zeb2^{Tg/Tg}* BM and spleen compared to Cre-only controls. Data are represented as mean + SD from 3 biological replicates/genotype except in (A) where only 2 wt controls were used. Male and female mice were analyzed ranging in age from 4 to 10 months of age. Raw data behind graphs and western blot are included in J of S1 and S2 Data, respectively. *$p < 0.05$; **$p < 0.01$, ***$p < 0.001$, nonparametric *t* test. BM, bone marrow; wt, wild-type.
(TIFF)

**S11 Fig. ZEB1 and ZEB2 expression and survival analysis in AML. (A)** Violin plots depicting normalized gene counts as TPM of ZEB1 (left) and ZEB2 (right) gene expression across AML samples aggregated by the presence of genomic gene fusions present in human AML [23]. Median including 75% and 25% quartiles are denoted from top to bottom as dashed lines. **(B)** Kaplan–Meier analysis of samples containing high (blue line) or low (red line) expression of ZEB1 (left) and ZEB2 (right). The appropriate cutoff was defined through a

scanning method implemented in the R2 database (http://r2.amc.nl). Bonferroni corrected *p*-values determine that higher ZEB1 statistically reduce the overall survival probability (**S9B Fig**, left), while higher expression of ZEB2 did not cause this effect. $^*p < 0.05$; $^{**}p < 0.01$, $^{***}p < 0.001$, Bonferroni-corrected nonparametric *t* test. **(C)** Increased expression of both ZEB1 (left) and ZEB2 (Right) appear to significantly correlate with increased numbers of leukemic blasts present in AML populations ($p = 0.0417$ and $p = 0.0024$, respectively). Raw data behind plots are included in **S7 Table**. AML, acute myeloid leukemia; TPM, transcripts per million.
(TIFF)

**S1 Text. Supporting information methods, materials, and references.**
(DOCX)

**S1 Table. Overall chimerism in BM donor competition experiments.** BM cells from *Zeb1* null CD45.2+ (Zeb1$^{Δ/Δ}$) mice or *Zeb1* heterozygous CD45.2+ cells (Zeb1$^{+/Δ}$) were mixed with equal numbers of control CD45.1+ BM cells and used to reconstitute lethally irradiated CD45.1+ mice. Numbers depict the percentage or donor and competitor cells in each experiment. BM, bone marrow.
(XLSX)

**S2 Table. EdgeRun DEGs lists obtained when comparing R26-Cre-ERT2; *Zeb2$^{fl/fl}$* (*Zeb2$^{Δ/Δ}$*), R26-Cre-ERT2; *Zeb$^{fl/+}$*, *Zeb1$^{fl/fl}$* (*Zeb2$^{Δ/+}$*, *Zeb1$^{Δ/Δ}$*) and *Zeb1$^{fl/fl}$* (*Zeb1$^{Δ/Δ}$*) against iDKO R26-Cre-ERT2; *Zeb1/2$^{fl/fl}$* (*Zeb1/2$^{Δ/Δ}$*), respectively, with a cutoff FDR <0.1.** We reported DEGs intersections between each comparison including the Z-score normalized counts across replicates. DEG, differentially expressed gene; FDR, false discovery rate; iDKO, inducible double knockout.
(XLSX)

**S3 Table. GO terms, KEGG pathway analysis and STRING gene networks analysis from the DEGs from Table 2.** We used a cutoff a FDR <0.05. FDR, false discovery rate; GO, gene ontology.
(XLSX)

**S4 Table. ChIP-seq analysis from ZEB1 and ZEB2 in GM12878 and K562 cell lines, respectively in 2 different ChIP-seq experiments (GEO accessions GSE32465 and GSE59395, respectively).** We calculated 1× normalized occupancy values in a window of 4000 base pairs around genes containing peaks that also are DEGs in the mouse (see **S1 Table**). Peak annotation was performed with the ChIPSeeker R package. ChIP-seq, Chromatin Immunoprecipitation Sequencing.
(XLSX)

**S5 Table. List of antibodies for cell cytometry and primers used in this study.**
(XLSX)

**S6 Table. Raw data used to make graphs and panels in S7 Fig.**
(XLSX)

**S7 Table. Raw data used to make graphs and panels in S11 Fig.**
(XLSX)

**S1 Data. Raw data used to make all graphs/panels in manuscript.**
(XLSX)

**S2 Data. Raw uncropped western blot.**
(PDF)

## Acknowledgments

We acknowledge that the CancerCare Manitoba Research Institute and the University of Manitoba campuses are located on the original lands of Anishinaabeg, Cree, Oji-Cree, Dakota, and Dene peoples and on the homeland of the Metis Nation.

## Author Contributions

**Conceptualization:** Jueqiong Wang, Carlos Farkas, Catherine Carmichael, Christian M. Nefzger, Jose M. Polo, Jody J. Haigh.

**Data curation:** Jueqiong Wang, Carlos Farkas, Aissa Benyoucef, Catherine Carmichael, Katharina Haigh, Nick Wong, Christian M. Nefzger, Steven Goossens.

**Formal analysis:** Jueqiong Wang, Carlos Farkas, Aissa Benyoucef, Catherine Carmichael, Nick Wong, Christian M. Nefzger, Jody J. Haigh.

**Funding acquisition:** Jody J. Haigh.

**Investigation:** Jueqiong Wang, Aissa Benyoucef, Catherine Carmichael, Katharina Haigh, Steven Goossens.

**Methodology:** Jueqiong Wang, Carlos Farkas, Katharina Haigh, Jose M. Polo.

**Project administration:** Jody J. Haigh.

**Resources:** Danny Huylebroeck, Marc P. Stemmler, Simone Brabletz, Thomas Brabletz, Geert Berx.

**Software:** Carlos Farkas, Nick Wong.

**Writing – original draft:** Carlos Farkas, Jody J. Haigh.

**Writing – review & editing:** Jueqiong Wang, Carlos Farkas, Aissa Benyoucef, Catherine Carmichael, Katharina Haigh, Nick Wong, Danny Huylebroeck, Marc P. Stemmler, Simone Brabletz, Thomas Brabletz, Steven Goossens, Geert Berx, Jose M. Polo, Jody J. Haigh.

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
