## [Editor Report · Decision Letter 0]

6 Dec 2020

Dear Dr Haigh, 

Thank you for submitting your manuscript entitled "Molecular interplay between the EMT transcription factors Zeb1 and Zeb2 in regulating hematopoietic stem and progenitor cell differentiation and their synergy in fine tuning hematopoietic lineage fidelity." for consideration as a Research Article by PLOS Biology.

Your manuscript has now been evaluated by the PLOS Biology editorial staff as well as by an academic editor with relevant expertise and I am writing to let you know that we would like to send your submission out for external peer review.

Please re-submit your manuscript within two working days, i.e. by Dec 08 2020 11:59PM.

Kind regards,

Ines

--

Ines Alvarez-Garcia, PhD,

Senior Editor

PLOS Biology

---

## [Decision Letter · Decision Letter 1]

10 Feb 2021

Dear Dr Haigh,

Thank you very much for submitting your manuscript entitled "Molecular interplay between the EMT transcription factors Zeb1 and Zeb2 in regulating hematopoietic stem and progenitor cell differentiation and their synergy in fine tuning hematopoietic lineage fidelity" for consideration as a Research Article at PLOS Biology. Thank you also for your patience as we completed our editorial process, and please accept my apologies for the delay in providing you with our decision. Your manuscript has been evaluated by the PLOS Biology editors, an Academic Editor with relevant expertise, and by two independent reviewers.

As you will see, both reviewers think the findings are interesting and complement recent papers on the topic, but they also ask for several clarifications, missing details from the experiments and several controls that should be added to strengthen the conclusions.

In light of the reviews (attached below), we will not be able to accept the current version of the manuscript, but we would welcome re-submission of a revised version that takes into account the reviewers' comments. We cannot make any decision about publication until we have seen the revised manuscript and your response to the reviewers' comments. Your revised manuscript is also likely to be sent for further evaluation by the reviewers.

We expect to receive your revised manuscript within 3 months. 

**IMPORTANT - SUBMITTING YOUR REVISION**

*Re-submission Checklist*

a) *Published Peer Review*

b) *PLOS Data Policy*

Sincerely,

Ines

--

Ines Alvarez-Garcia, PhD,

Senior Editor,

PLOS Biology

Reviewers’ comments

Rev. 1:

Summary:

The authors are interested in the role of transcription factors ZEB1 and ZEB2 in hematopoietic stem cell (HSC) maintenance and differentiation. The authors use a number of complex model systems to look at knockout of Zeb1 and Zeb2, along with overexpression of the genes. It is taken under consideration the time and work to perform the complex breeding. The studies define the role of Zeb1 and Zeb2 in hematopoietic stem and progenitor populations along with lineage differentiation. In addition, the authors explore the role of the genes in acute myeloid leukemia (AML) progression. Overall, the manuscript contains substantial amount of data and findings on Zeb1, and Zeb2, however important details are missing from the manuscript that would strengthen the conclusions. Unfortunately, lack of detail in portions of the manuscript, poor figure quality, and conclusions not supported by the data greatly reduces the enthusiasm. I would recommend major revisions of the manuscript.

Weaknesses:

* In the introduction the authors mention that ZEB1 and ZEB2 potentially contribute to development and/or maintenance of leukemia. A number of large data sets are available that look at expression and survival in AML. It would strengthen the manuscript to include expression data, especially in MLL subset of patients. Are ZEB1 and ZEB2 highly expressed in AML and is expression correlated with poor survival?

* What is the expression of Zeb1 and Zeb2 in hematopoietic lineages? Figure 1 demonstrates stem and progenitor populations, but it is not mentioned in the manuscript more mature populations. This data should be summarized and referenced, if previously published.

* Figure 1A is very difficult to interpret, and would benefit from being displayed differently (ex; violin plot). It is unclear what the cell types indicated on the x-axis represent. What is meant by projected, broad, ect.?

* In Figure 2, the authors display data for myeloid populations, however lymphoid populations are not mentioned. Also, the SLAM markers (CD48 and CD150) are the standard markers for HSC populations and should be performed in parallel with CD34 and CD135.

* On page 8, the authors look at CMP population, which is listed as LK CD34+ CD16/32-. This is inaccurate since the CMP population is CD16/32 low. The authors should include a gating strategy for the CMP, MEP, and GMP populations.

* Also, in page 8 the authors say "Overall, these results are in contrast to Zeb2 deficient hematopoietic null mice that showed enhanced granulocytic differentiation, decreased GMPs…." Is this completely in contrast? In Figure 2C, the bar graph shows a decrease of GMP. In addition, the LSK population contains the LT-HSC, ST-HSC, and MPP, however the data displayed for % live cells depicts more LT-HSC and ST-HSC than total LSK.

* Figure 3C, what is the overall chimerism, and what is the representation of each lineage?

* On page 15, the authors state "Overall, we have discovered that many genes may be co-repressed by Zeb1 and Zeb2…." RNAseq data alone does not support this conclusion. This would need validation.

* For Figure 7, a number of details are missing, including time-points, age, significance, impact on survival? Also, it would be beneficial to include protein expression along with GFP expression. Is data significant with only 3 representative mice?

* Experimental details are lacking for Figure 8. Was only one primary donor used for the cohort? Was the phenotype of the primary mice identical? What was the tumor burden similar? Were these leukemia cells from peripheral blood, spleen or bone marrow?

Minor Weaknesses:

* Manuscript needs attention to proper gene and protein nomenclature.

* A number of figures that font is too small to be legible (ex; Figure 6, Figure 5C)

* The gating strategy is unclear in Figure 2, are numbers based on gating of CD45.2+ cells?

* Numbers in Figure 3A are confusing. Why in 1st plating is there 8 and 7 but in secondary 4 and 6, but at the end of the figure legend it says 4 biological replicates?

* Figure 5 key doesn't match the text

Rev. 2:

In this manuscript, the authors investigate the role of Zeb1 and Zeb2 in hematopoiesis, using conditional mouse knockout and overexpression models. A number of other investigators have published related articles, and therefore the authors focused on the effects of Zeb1 and double Zeb1/Zeb2 knockouts. The provide data on Zeb1 and Zeb2 expression, the effects of deletion, rescue by Zeb2, assessment of effects on downstream genes following deletion, overexpression, and effect of loss of Zeb1/2 on an AML model. In general, the experiments complement studies from other laboratories, and add to our knowledge of the role of Zeb1/2 in hematopoiesis and leukemia. This reviewer has a number of suggestions for improvement of the manuscript.

(1) In general, the authors utilize fetal liver cells rather than adult bone marrow in analyzing the effects of loss of Zeb1, perhaps because of the study of Almotiri et al. They should discuss how use of fetal versus adult cells may or may not lead to differences, given that fetal HSCs are proliferative and adult relatively non-proliferative. Can this account for any difference with the study of Almotiri?

(2) Similarly, the authors should discuss potential differences between fetal liver cell transplants into adult recipients (Fig. 2) versus using adult bone marrow as donor.

(3) The replating assays (Fig. 3A) were performed with fetal liver, and only for 2 replatings. If the authors have to analyze fetal and liver and/or adult bone marrow from Zeb1 KO mice for other experiments, they should repeat the fetal liver and compare with adult bone marrow (continuing, for at least the adult, for 3-4 rounds. However, this reviewer would not insist on breeding mice solely for this purpose.

(4) While the authors are utilizing mouse strains described in previous publications, this reviewer would like to see more information about the knockouts without having to look up a another paper. It is suggested to provide a diagram and some information about the Zeb1 and Zeb2 knockout strategy. In addition, the authors should provide information in the supplement showing loss of RNA and protein in the knockouts. Was any residual non-deleted RNA or protein detected?

(5) A number of investigators have demonstrated data in both murine and human models that some degree of myeloid differentiation is essential for leukemic transformation in AML. The authors should discuss whether this might have some relationship to the enhanced survival observed in the MLL-AF9 model?

---

## [Decision Letter · Decision Letter 2]

2 Aug 2021

Dear Dr Haigh,

Thank you for submitting your revised Research Article entitled "Molecular interplay between the EMT transcription factors ZEB1 and ZEB2 in regulating hematopoietic stem and progenitor cell differentiation and their synergy in fine tuning hematopoietic lineage fidelity." for publication in PLOS Biology. I have now obtained advice from the two original reviewers and have discussed their comments with the Academic Editor. 

Based on the reviews, we will probably accept this manuscript for publication, provided you satisfactorily address the data and other policy-related requests (see requests below my signature).

In addition, we would encourage you to consider the comment made by Reviewer 2 regarding whether you have RNA and/or protein data available.

We would also like you to consider a suggestion to improve the accessibility of the title, which is currently quite long and convoluted:

"The EMT transcription factors ZEB1 and ZEB2 interact to regulate hematopoietic stem and progenitor cell differentiation and hematopoietic lineage fidelity"

or

"Interplay between the EMT transcription factors ZEB1 and ZEB2 to regulate hematopoietic stem and progenitor cell differentiation and hematopoietic lineage fidelity"

We expect to receive your revised manuscript within two weeks. 

*Published Peer Review History*

*Early Version*

Sincerely,

Ines

--

Ines Alvarez-Garcia, PhD,

Senior Editor,

ialvarez-garcia@plos.org,

PLOS Biology

ETHICS STATEMENT:

Thank you for providing the ethics statement. Please also include the approval/license number.

DATA POLICY: IMPORTANT, PLEASE READ

Fig. 2C-F; Fig. 3A, C; Fig. 4B, C; Fig. 5A, B; Fig. 6B; Fig. 7A, C; Fig. 8B, C, E, F, H; Fig. 9B, C; Fig. S1; Fig. S3A-C, E; Fig. S4A, B; Fig. S5C, E; Fig. S6A-E; Fig. S7A, C, D; Fig. S8A; Fig. S9B, C, E and Fig. S10A-C

Please also ensure that figure legends in your manuscript include information on WHERE THE UNDERLYING DATA CAN BE FOUND, and ensure your supplemental data file/s has a legend. Please ensure as well that your Data Statement in the submission system accurately describes where your data can be found.

For figures containing FACS data, we ask that you provide FCS files and a picture showing the successive plots and gates that were applied to the FCS files to generate the figure. We can recommend you to deposit the data in the FlowRepository (http://flowrepository.org/) database, and please remember to make the files publicly available at this stage.

We require the original, uncropped and minimally adjusted images supporting all blot and gel results reported in an article's figures or Supporting Information files. We will require these files before a manuscript can be accepted so please prepare and upload them now. Please carefully read our guidelines for how to prepare and upload this data: https://journals.plos.org/plosbiology/s/figures#loc-blot-and-gel-reporting-requirements 

Reviewers' comments

Rev. 1:

Accept

Rev. 2:

 In general, the authors have done a very good job addressing the reviewer's comments. This reviewer has only two comments:

 One comment was the suggestion to use proper gene and protein nomenclature, which the authors proceeded to do in the revised manuscript. Although the authors have to do this to satisfy the "Nomenclature Police", this reviewer would like to make the point that HUGO rules can sometimes be very useful (as in the case of RUNX family nomenclature), and sometimes be completely idiotic (do you know what spfi1 is? Is it PU.1 or SP1, they are VERY DIFFERENT transcription factors, and HUGO nomenclature is causing great confusion, not clarification!). We scientists should not have to be slaves to people who are ignorant of what they are doing, but unfortunately we are. When this reviewer is Emperor, this will change.

 A second comment was "In addition, the authors should provide information in the supplement showing loss of RNA and protein in the knockouts. Was any residual non-deleted RNA or protein detected?" The authors commented on a residual floxed allele, which represents DNA, not RNA or protein. If the authors have RNA and/or protein data available, it would be good to include it as part of a supplementary figure, but this is a relatively minor point and this reviewer would not insist on additional experimentation for this.

---

## [Editor Report · Decision Letter 3]

20 Aug 2021

Dear Dr Haigh,

On behalf of my colleagues and the Academic Editor, Connie Eaves, I am pleased to say that we can in principle offer to publish your Research Article entitled "Interplay between the EMT transcription factors ZEB1 and ZEB2 regulates hematopoietic stem and progenitor cell differentiation and hematopoietic lineage fidelity." in PLOS Biology, provided you address any remaining formatting and reporting issues. These will be detailed in an email that will follow this letter and that you will usually receive within 2-3 business days, during which time no action is required from you. Please note that we will not be able to formally accept your manuscript and schedule it for publication until you have made the required changes.

PRESS

Sincerely, 

Ines

--

Ines Alvarez-Garcia, PhD 

Senior Editor 

PLOS Biology
